# SceneDirector: Bridging Explicit Geometry and Generative Priors for Unified Driving Scene Editing

**Yiyuan Liang** [1 2]   **Zhiying Yan** [1 2]   **Tao Zhang** [1 2]   **Shangke Liu** [3]   **Kai Lin** [1]
**Xu Zou** [1 2]   **Nong Sang** [1 2]   **Changxin Gao** [1 2]

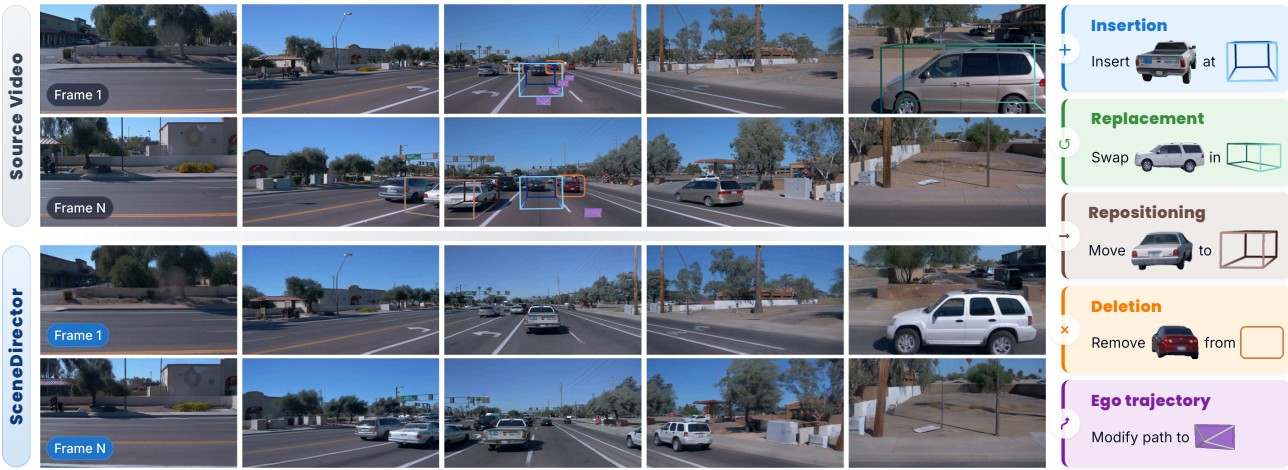

*Figure 1.* SceneDirector is a unified framework for multi-view driving video editing. It enables simultaneous 3D box-defined **object editing** (insertion, deletion, replacement, repositioning) and **ego-trajectory editing** in a single inference pass. By bridging geometric guidance with generative priors, it reconciles photorealistic synthesis for object editing and structural consistency for trajectory control.

## Abstract

Validating autonomous driving systems requires diverse scenarios, yet real-world data collection is biased and costly. Editing existing driving logs offers a scalable solution, but simultaneously editing objects and the ego-trajectory—termed unified editing—remains challenging. Current methods face an inherent dilemma: generative flexibility for object editing and physical precision for trajectory control. To address this, we introduce SceneDirector, a diffusion-based framework that bridges explicit geometry and generative priors. For explicit geometry, we leverage LiDAR-guided depth completion to construct dense scene geometry and integrate editable 3D assets to form a Unified Geometric Scaffold, providing rigorous structural guidance for unified editing. To leverage generative priors, we encode the source video into a Static Texture Bank to provide rich appearance context. Our proposed Mask-Gated Reference Attention bridges these modalities. Guided by a geometric uncertainty metric, this mechanism dynamically regulates the interaction between the scaffold and the bank—preserving reliable geometry while adaptively injecting textures for semantic refinement. Extensive evaluations demonstrate that SceneDirector outperforms state-of-the-art methods in both controllability and visual quality.

## 1. Introduction

Autonomous driving (AD) systems require rigorous validation across a wide range of diverse scenarios to ensure robustness in real-world deployment. However, real-world data collection is inherently biased toward nominal driving conditions, while safety-critical cases remain rare and expensive to capture at scale. Editing existing driving logs offers a scalable alternative for synthesizing diverse training

[1]School of Artificial Intelligence and Automation, Huazhong University of Science and Technology, Wuhan, China [2]State Key Laboratory of Multispectral Information Intelligent Processing Technology, Wuhan, China [3]Cornell University, Ithaca, NY, USA. Correspondence to: Changxin Gao <cgao@hust.edu.cn>.

*Proceedings of the 43rd International Conference on Machine Learning*, Seoul, South Korea. PMLR 306, 2026. Copyright 2026 by the author(s).

samples while maintaining high fidelity. Simultaneous control over local object manipulation and global ego-trajectory editing offers comprehensive flexibility for simulation, as it integrates these distinct edits into a cohesive system, thereby enabling reactive scenario generation.

However, unifying these tasks necessitates reconciling two conflicting requirements: generative flexibility for object editing and physical precision for trajectory control. On the one hand, generating high-fidelity details such as object shadows, lighting changes, and occlusion completion requires powerful generative priors (Hassan et al., 2025; Wang et al., 2025). However, without strict spatial constraints, this generative freedom compromises 3D alignment, causing geometric drift under viewpoint shifts. On the other hand, ensuring view consistency under ego-trajectory changes demands explicit 3D geometry (Chen et al., 2025b; Yan et al., 2024). Conversely, lacking generative capacity, these rigid representations cannot plausibly fill occluded regions or match lighting, leading to artifacts during object editing. While recent works (Zhao et al., 2025a; Yan et al., 2025) incorporate diffusion into reconstruction pipelines, they restrict it to texture refinement rather than semantic creation for substantial edits. Thus, reconciling the structural rigor required for trajectory control with the semantic flexibility required for object editing remains challenging.

To address this, we introduce SceneDirector, a framework that bridges explicit geometry and generative priors. Specifically, to establish explicit geometry, we construct a Unified Geometric Scaffold by fusing LiDAR-guided completed depth with editable 3D assets. Eliminating the need for per-scene training, this explicit structure is computationally efficient and guarantees structural fidelity while naturally accommodating 3D asset integration, thereby serving as a unified guide for both object and trajectory control. To leverage generative priors, we formulate the source video as a Static Texture Bank, encoding it into a static key-value memory to provide rich appearance context. Finally, we propose Mask-Gated Reference Attention to bridge these paradigms. It uses geometric features from the scaffold as spatial queries to attend to the Texture Bank, while using an uncertainty metric to discern reliable geometry from unavoidable occlusions in the scaffold. This gating preserves the reliable structural layout and dynamically regulates texture injection to enhance photorealism in semantically complex regions.

SceneDirector enables simultaneous object editing (insertion, deletion, replacement, repositioning) and ego-trajectory editing within a single inference pass. For object editing, it is agnostic to asset provenance, integrating diverse inputs ranging from scanned datasets (Du et al., 2025) to generative assets synthesized via text-to-3D or single-image reconstruction pipelines (e.g., Wu et al., 2025; Xiang et al., 2025). For ego-trajectory editing, it enables free-form view-point control while harmonizing geometric precision with generative fidelity. Our main contributions are as follows:

- We propose SceneDirector, a system-level framework for unified object and ego-trajectory editing. By bridging explicit geometry and generative priors, it reconciles structural consistency for trajectory control and photorealistic synthesis for object manipulation.

- We achieve this bridge via a Unified Geometric Scaffold for structural guidance and a Static Texture Bank for appearance context, integrated by Mask-Gated Reference Attention that preserves reliable structure while synthesizing details in semantically complex regions.

- We demonstrate SceneDirector's superior controllability and visual fidelity through extensive experiments.

## 2. Related Work

### 2.1. Driving Scene Generation

The evolution of driving scene generation has shifted from structurally conditioned synthesis to predictive world modeling with granular control. Early works (Li et al., 2025a; Gao et al., 2024b;a; Russell et al., 2025; Zhao et al., 2025b) predominantly focused on ensuring spatial consistency by conditioning on explicit 3D priors, such as semantic occupancy grids, HD maps, and bounding boxes. Building upon this, the focus shifted toward predictive world models (Wang et al., 2024b; Zheng et al., 2024; Gao et al., 2024c) and foundation models (Ren et al., 2025), which are designed to forecast future states while prioritizing temporal coherence and video fidelity. To further broaden the functional scope, Orbis (Mousakhan et al., 2025) and Vista (Gao et al., 2024c) address the stability of long-horizon dynamics, whereas DriVerse (Li et al., 2025c) and GEM (Hassan et al., 2025) propose flexible conditioning paradigms using multimodal trajectory prompts and object-centric visual embeddings to facilitate fine-grained manipulation of scene evolution.

### 2.2. Driving Scene Editing

Methodologies for driving scene editing are generally categorized into local object manipulation and global trajectory synthesis. Composition-based approaches (Chen et al., 2021; Li et al., 2022; Bai et al., 2024) represent an early paradigm, inserting objects via explicit geometric rendering, though often struggling with photorealistic composition. The advent of diffusion models has shifted the focus to generative object editing, where works such as DriveEditor (Liang et al., 2025b), GenMM (Singh et al., 2024), and SceneCrafter (Zhu et al., 2025) leverage 3D layout conditions for spatially grounded synthesis. To further enhance geometric fidelity, $G^2$Editor (Li et al., 2025b) and R3D2 (Ljungbergh et al., 2025) integrate reusable 3D assets

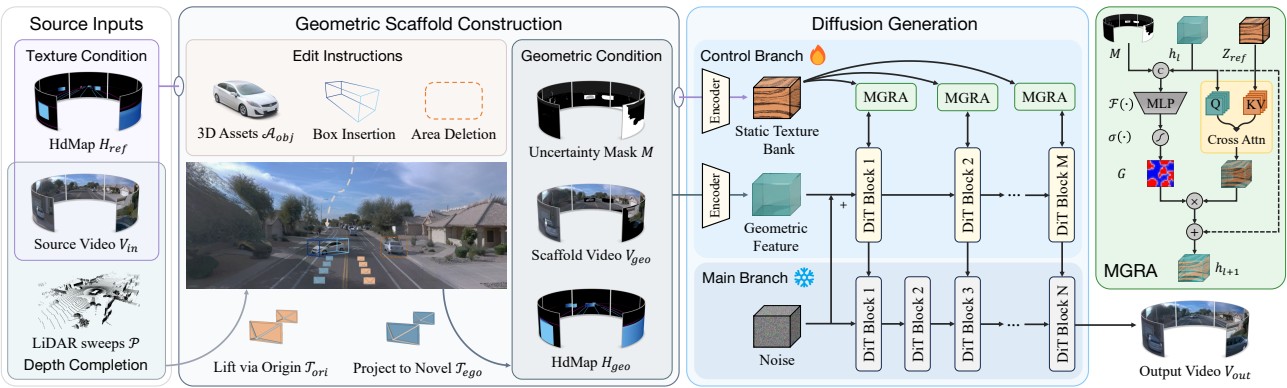

*Figure 2.* Overview of SceneDirector. **Left (Unified Geometric Scaffold Construction):** The scaffold is constructed by fusing LiDAR-guided depth with editable 3D assets. It is then rendered under the target ego-trajectory $\mathcal{T}_{ego}$ to yield the Geometric Video $V_{geo}$ (structural guide) and Uncertainty Mask $M$ (reliability metric). **Right (Diffusion Generation):** The model synthesizes the output video $V_{out}$ using a dual-branch DiT. The texture condition is encoded into a Static Texture Bank to provide appearance context. We introduce Mask-Gated Reference Attention (MGRA) to bridge explicit geometry and generative priors. It utilizes $M$ to dynamically regulate the interaction between the scaffold and the Texture Bank—preserving reliable geometry while adaptively injecting textures for semantic refinement.

or relightable priors into the diffusion process. In parallel, trajectory-centric methods aim to tackle consistency under ego-motion changes. GeoDrive (Chen et al., 2025a) and FreeVS (Wang et al., 2025) utilize geometry-aware warping for novel view synthesis, while DiST-4D (Guo et al., 2025) and Stag-1 (Wang et al., 2024a) explore disentangled spatio-temporal representations to maintain video coherence. However, a unified framework capable of simultaneously handling object editing and ego-trajectory editing within a single inference pass remains an open challenge.

### 2.3. Driving Scene Reconstruction

Reconstruction pipelines (Yang et al., 2023; Chen et al., 2025b; Yan et al., 2024; Huang et al., 2024a) decompose scenes into static backgrounds and dynamic actors using neural representations (e.g., NeRF (Mildenhall et al., 2021), 3DGS (Kerbl et al., 2023)) for replay. Some works (Wei et al., 2024; Xiong et al., 2025; Lu et al., 2025) provide limited scene editing capabilities but lack visual authenticity. Nevertheless, these explicit methods are fundamentally limited by input coverage and lack the generative capacity to plausibly synthesize occluded regions exposed by static object removal or significant trajectory deviations. To address rendering artifacts in novel views, recent hybrid frameworks (Zhao et al., 2025a; Yan et al., 2025; Ni et al., 2025; Mao et al., 2025; Lin et al., 2025; Chen & Peng, 2025) have augmented 3D representations with video diffusion priors. Crucially, these generative components are primarily optimized for visual enhancement rather than semantic manipulation, focusing on texture refinement rather than inferring content for complex counterfactual edits. In contrast, our approach uses an efficient, training-free geometric scaffold to actively guide the generative model, enabling it to synthesize coherent details for substantial scene alterations.

## 3. Method

Formally, SceneDirector models the conditional generation of a driving video $V_{out}$ given a source video $V_{in}$, LiDAR sequence $\mathcal{P} = \{P_t\}_{t=1}^{T}$, and a set of user-defined edits. Specifically, object manipulation is defined by target 3D bounding boxes $\mathcal{B}_{obj} \in \mathbb{R}^{T \times N \times 8 \times 3}$ with associated 3D assets $\mathcal{A}_{obj}$, while trajectory control is governed by a target ego-pose sequence $\mathcal{T}_{ego} \in \mathbb{R}^{T \times 4 \times 4}$. Sec. 3.1 first outlines the Diffusion Transformer backbone. Sec. 3.2 details the training data construction pipeline. Sec. 3.3 details the construction of the Unified Geometric Scaffold, which integrates depth-completed point clouds and 3D assets into a cohesive spatial representation to enforce structural layout. Finally, Sec. 3.4 introduces our Mask-Gated Reference Attention, which preserves reliable structure while dynamically regulating texture injection to synthesize details.

### 3.1. Preliminaries

Our work builds upon Cosmos-Transfer2.5 (Ali et al., 2025), a state-of-the-art video world model based on Diffusion Transformers (DiT; Peebles & Xie, 2023). Input videos are first compressed into continuous latents $x$ using the Causal 3D VAE in WAN2.1 (Wan et al., 2025). Unlike standard diffusion, the model adopts a Rectified Flow (Liu et al., 2023) objective, defining the forward process as a linear interpolation $x_t = (1 - t)x + t\epsilon$ between data $x$ and noise $\epsilon \sim \mathcal{N}(0, I)$, where $t = 0$ denotes clean data and $t = 1$ denotes pure noise. The denoising network $v_\theta$ learns to predict the constant velocity field $u_t = \epsilon - x$ by minimizing:

$$\mathcal{L}(\theta) = \mathbb{E}_{t,x,\epsilon,\mathbf{c}} \left[ \| v_\theta(x_t, t, \mathbf{c}) - (\epsilon - x) \|^2 \right], \quad (1)$$

where $\mathbf{c}$ encompasses the set of conditioning signals. Specifically, the model is conditioned on text descriptions $c_t$ en-

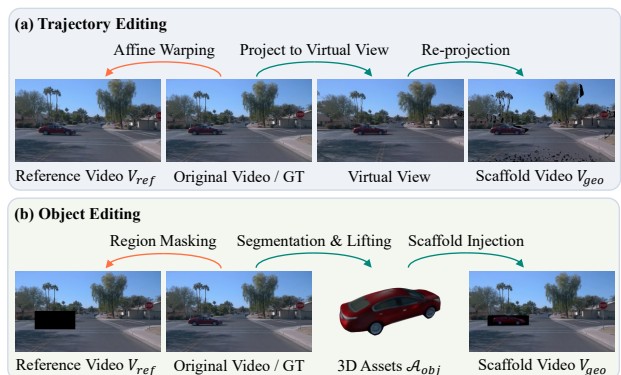

*Figure 3.* Pipeline for self-supervised pair construction. **(a) Trajectory:** We synthesize a misaligned $V_{ref}$ through warping, and generate $V_{geo}$ via round-trip projection to mimic sparsity artifacts. **(b) Object:** We mask objects in $V_{ref}$ to prevent shortcut learning, while composing lifted 3D assets into $V_{geo}$ for structural guidance.

coded by a Vision-Language Model (Azzolini et al., 2025) and spatial controls $c_s$. We extend the architecture to bridge explicit geometric guidance and generative priors via our proposed Mask-Gated Reference Attention mechanism.

## 3.2. Self-Supervised Pair Construction

A core challenge in training video editing models is the absence of paired data, i.e., ground-truth videos corresponding to novel trajectories or modified objects. When such real paired supervision is unavailable, a common self-supervised degradation paradigm is to synthesize pseudo pairs, as adopted in Monodepth2 (Godard et al., 2019), FreeVS (Wang et al., 2025), and FreeSim (Fan et al., 2025). Following this paradigm, we formulate a self-supervised reconstruction task. We treat the original video clip $V_{gt}$ as the reconstruction target and synthesize training triplets to simulate the geometric and appearance discrepancies encountered during inference.

**Synthetic View Perturbation.** To mimic the spatial misalignment encountered in trajectory editing, we synthesize a structurally misaligned reference video $V_{ref}$ and a degraded yet spatially aligned scaffold video $V_{geo}$. As shown in Figure 3 (a), we first define a perturbation function $\Phi$ (detailed in the Appendix D) to simulate ego-motion discrepancies. Specifically, $\Phi$ covers forward, backward, left, and right trajectory deviations: forward/backward motion is approximated by temporally resampling neighboring frames, while left/right deviations are modeled by smooth lateral motion together with a coupled yaw change. The image warping is implemented in a view-dependent manner: we use affine warping for front-facing cameras and perspective homography for side-view cameras to approximate parallax under viewpoint changes. Applying $\Phi$ to the ground-truth video yields the misaligned reference video, $V_{ref} = \Phi(V_{gt})$.

Second, to generate the aligned scaffold video $V_{geo}$, we ap-

ply the same perturbation to the geometric scaffold through a round-trip projection process, which mimics the sparsity and reprojection artifacts encountered at inference time. The point cloud is first rendered according to $\Phi$ to obtain intermediate RGB and depth maps. These maps are then lifted back into 3D space and reprojected to the recorded ego-pose. Finally, we rasterize the projected points to obtain the scaffold video $V_{geo}$ and the Uncertainty Mask $\mathbf{M}$.

**3D Asset Curation.** To facilitate object-centric editing, we build a library of high-fidelity 3D assets. We construct an automated filtering pipeline (detailed in the Appendix D) to select objects with sufficient visibility and temporal stability. Valid observations are segmented via SAM2 (Ravi et al., 2025), verified for semantic integrity using Qwen2.5-VL (Bai et al., 2025), and lifted into 3D assets $\mathcal{A}_{obj}$ using Trellis (Xiang et al., 2025). To prevent shortcut learning, we mask the projected regions of $\mathcal{B}_{obj}$ in $V_{ref}$. Once curated, these assets serve as the source material for both the training and inference pipelines.

**Training Pairs Formulation.** We formulate the training set as a collection of samples $\mathcal{D} = \{(\mathcal{S}^{(i)}, \mathcal{C}^{(i)}, V_{gt}^{(i)})\}_{i=1}^{N_{data}}$. Specifically, $\mathcal{S} = (V_{geo}, \mathbf{M}, H_{geo})$ denotes the geometric condition, serving as the spatially-aligned structural anchor, while $\mathcal{C} = (V_{ref}, H_{ref})$ represents the texture condition, encapsulating the misaligned appearance cues. $H_{geo}$ and $H_{ref}$ denote the HDMaps corresponding to the recorded and perturbed trajectories, respectively. The model is optimized to reconstruct the ground truth $V_{gt}$ by querying texture cues from $\mathcal{C}$ and aligning them with the layout dictated by $\mathcal{S}$.

## 3.3. Unified Geometric Scaffold Construction

To enable precise manipulation of the ego-trajectory and specified target objects, we explicitly construct the Unified Geometric Scaffold, a 3D composition that fuses the preserved scene context with manipulable object assets $\mathcal{A}_{obj}$, and render it under the target ego-trajectory $\mathcal{T}_{ego}$ to yield a coherent structural guidance sequence.

**Scene Composition and Editing.** We formulate the scene at each time step $t$ as a composite of the context $\mathcal{P}_{ctx}$ (encompassing the static background and non-edited entities) and the set of target 3D assets $\mathcal{A}_{obj}$. We build dense scene geometry via a multi-modal depth completion network (DMD³C; Liang et al., 2025a), fusing the RGB image $I_t$ with raw LiDAR sweeps to predict a depth map $D_t$. These 2D observations are then lifted into the global 3D space. Formally, a pixel $\mathbf{u} = (u, v)$ with predicted depth $d = D_t(\mathbf{u})$ is back-projected to its world coordinate $\mathbf{x}$ via:

$$\mathbf{x} = \mathbf{R}_t \cdot \left( d \cdot \mathbf{K}_{cam}^{-1}[u, v, 1]^{\top} \right) + \mathbf{t}_t, \tag{2}$$

where $\mathbf{K}_{cam}$ is the camera intrinsic matrix. $\mathbf{T}_t = \{\mathbf{R}_t, \mathbf{t}_t\}$ denotes the ego-pose, which transforms points from the local camera frame to the global world frame.

**Projective Rendering under Novel Trajectories.** Given a novel ego-trajectory $\mathcal{T}_{ego} = \{\mathbf{T}'_t\}_{t=1}^T$, where each $\mathbf{T}'_t \in SE(3)$ represents the target ego-pose, we render the point cloud $\mathcal{P}_t$ into a sequence of geometric maps. For an arbitrary point $\mathbf{x}_i \in \mathcal{P}_t$, its projected image-plane coordinate $\mathbf{u}_{i,t} = [u, v]^\top$ is governed by the pinhole camera model:

$$s \cdot [u, v, 1]^\top = \mathbf{K}_{cam} \cdot [\mathbf{I}|\mathbf{0}] \cdot (\mathbf{T}'_t)^{-1} \cdot [\mathbf{x}_i^\top, 1]^\top, \quad (3)$$

where $s$ is the scale factor representing depth. We rasterize the composite point cloud into the scaffold video $V_{geo}$, employing a Z-buffer mechanism to resolve visibility conflicts by retaining the color of the nearest surface at each pixel.

**Sparsity and Uncertainty Modeling.** Rendering from novel trajectories inevitably introduces disocclusion holes where geometry is completely undefined. Conversely, regions beyond the sensor's field-of-view (e.g., sky) rely entirely on depth completion. While these filled regions lack the precision of raw measurements, they still provide essential low-frequency structural cues. To explicitly quantify the reliability of these geometric cues, we assign a discrete uncertainty category index $\mathbf{M}_t(\mathbf{u}) \in \{0, 1, 2, 3\}$ based on the source of the surface point projected to each pixel. Let $\mathbf{x_u}$ denote the nearest 3D surface point visible at pixel $\mathbf{u}$:

$$\mathbf{M}_t(\mathbf{u}) = \begin{cases} 0 & \text{if } \mathbf{x_u} \in \mathcal{P} & \text{(Sensor-Verified)} \\ 1 & \text{if } \mathbf{x_u} \in \mathcal{P}_{ctx} \setminus \mathcal{P} & \text{(Inferred Layout)} \\ 2 & \text{if } \mathbf{x_u} \in \mathcal{A}_{obj} & \text{(Synthetic Asset)} \\ 3 & \text{otherwise} & \text{(Voids)} \end{cases} \quad (4)$$

By distinguishing precise sensor measurements from inferred structures, the mask explicitly categorizes the reliability of the geometric cues. This guidance serves as the core condition for our Mask-Gated Reference Attention.

### 3.4. Mask-Gated Reference Attention

The Unified Geometric Scaffold enforces rigorous spatial alignment but lacks texture in sensor-denied regions. Conversely, the source video provides rich details yet suffers from spatial misalignment. To bridge this gap, we introduce Mask-Gated Reference Attention (MGRA), an uncertainty-aware mechanism that dynamically regulates texture injection based on local geometric reliability.

We instantiate the Static Texture Bank by encoding the texture condition $\mathcal{C} = (V_{ref}, H_{ref})$ into a frozen latent representation $\mathbf{Z}_{ref}$. To ensure computational efficiency, this representation is shared across all blocks, serving as a static memory queried by the geometric features derived from $\mathcal{S} = (V_{geo}, \mathbf{M}, H_{geo})$. Formally, in each control block $l$, the intermediate geometric features $\mathbf{h}_l$ serve as the query $\mathbf{Q}_l = \mathbf{h}_l \mathbf{W}_Q$. This query attends to the keys and values $\mathbf{K}, \mathbf{V}$ projected from $\mathbf{Z}_{ref}$, retrieving corresponding textures from the global context despite spatial misalignment.

However, naive attention injection is suboptimal: it risks overwriting precise structural cues in geometrically verified regions while failing to hallucinate details in occluded areas. To address this, we formulate the fusion process as a reliability-aware gating mechanism. We introduce a learned reliability gate $\mathbf{G}$ that modulates the trade-off between the fidelity of the local geometric signal and the richness of the reference bank, explicitly guided by the sensor's epistemic uncertainty. We define this gating function as:

$$\mathbf{G} = \sigma\left(\mathcal{F}\left([\mathbf{h}_l; \Psi(\mathbf{M})]\right)\right), \quad (5)$$

where $[\cdot; \cdot]$ denotes channel concatenation, $\sigma$ is the Sigmoid activation, and $\mathcal{F}$ is a multilayer perceptron (MLP). $\Psi(\cdot)$ denotes an embedding operator that downsamples $\mathbf{M}$ to match the latent resolution and projects the category indices into a continuous feature space using a learnable embedding layer. Jointly conditioning on $\mathbf{M}$ and $\mathbf{h}_l$ allows the gate to be content-adaptive, enabling the mechanism to discern semantic complexity even where sensor coverage is uniform.

The retrieved texture information is then injected into the control stream via a modulated residual connection. The feature update process is formulated as:

$$\mathbf{h}_{l+1} = \mathbf{h}_l + \lambda \cdot \left(\mathbf{G} \odot \text{Attention}(\mathbf{Q}_l, \mathbf{K}, \mathbf{V})\right), \quad (6)$$

where $\odot$ represents element-wise multiplication and $\lambda$ is a zero-initialized learnable scaling factor. This mechanism naturally induces a spatially adaptive disentanglement: the gate suppresses the reference stream ($\mathbf{G} \to 0$) in structurally homogeneous regions to strictly enforce geometric constraints, while promoting it ($\mathbf{G} \to 1$) in semantically rich regions to facilitate texture synthesis from the bank.

## 4. Experiments

### 4.1. Object editing

**Evaluation Benchmark.** To benchmark object-editing capabilities, we curate 64 representative evaluation scenarios with diverse object types and scene layouts from the Waymo Open Dataset (WOD; Sun et al., 2020) validation split via a fully automated pipeline (detailed in Appendix C). We establish three protocols: **Single-Edit** (insertion) isolates geometric precision and fidelity from sequential errors. **Multi-Edit** evaluates the ability to synthesize complex scenarios requiring simultaneous multi-type editing. **Unified Capability Analysis** validates joint object-and-trajectory editing by executing object manipulation simultaneously with ego-trajectory modification (2m Gradual Transition).

**Baselines and Protocol.** We compare SceneDirector against two object-editing baselines: *VACE-14B* (Jiang et al., 2025), a large-scale general-purpose video editor that integrates multiple tasks under a unified conditioning interface; and

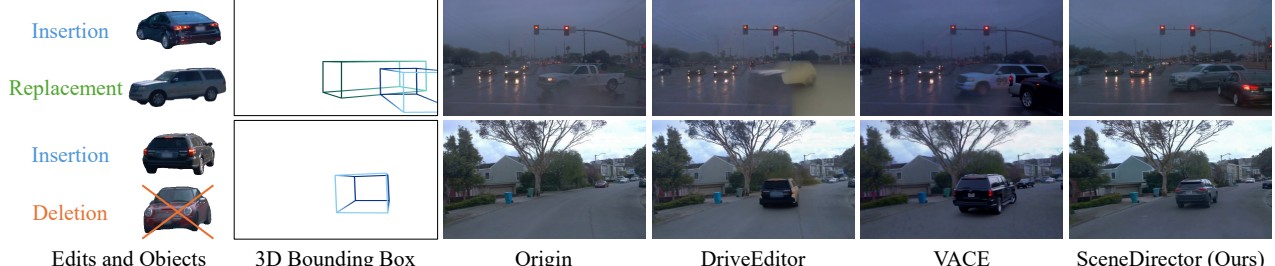

Edits and Objects  3D Bounding Box  Origin  DriveEditor  VACE  SceneDirector (Ours)

*Figure 4.* Qualitative comparison on multi-object editing. Each row illustrates a scenario combining two distinct editing operations simultaneously. **Left to Right:** Original frames; target 3D assets (crossed-out items denote deletion); and 3D layout conditions (Blue: Insertion, Green: Replacement). **Observations:** *DriveEditor* suffers from residual artifacts (e.g., gray patches) due to ineffective masking strategies. *VACE* struggles with precise geometric alignment and may erroneously hallucinate objects during target deletion. *SceneDirector* synthesizes high-fidelity assets that are strictly aligned with the 3D bounding boxes and harmoniously integrated into the scene context.

*Table 1.* Quantitative comparison on object editing. **Multi-Edit** evaluates complex scenarios requiring multiple object manipulations, where SceneDirector performs all edits jointly in a single inference pass. **Single-Edit** isolates intrinsic generation quality and spatial controllability, showing that our method achieves the lowest ATE/AOE while maintaining competitive visual quality. The bottom rows further assess the multi-view setting: simultaneous object and trajectory editing (*Obj.+Traj.*) introduces only marginal degradation compared with object-only editing (*Obj. Only*), demonstrating that SceneDirector preserves precise object control under unified editing.

| METHOD | MULTI-EDIT | | | | | SINGLE-EDIT | | | | |
|---|---|---|---|---|---|---|---|---|---|---|
| | FID ↓ | FVD ↓ | CLIP-I ↑ | ATE ↓ | AOE ↓ | FID ↓ | FVD ↓ | CLIP-I ↑ | ATE ↓ | AOE ↓ |
| VACE | 49.18 | 729.42 | 76.36 | 1.09 | 0.077 | **35.04** | **451.31** | 76.98 | 1.02 | 0.075 |
| DRIVEEDITOR | 51.43 | 818.48 | 75.98 | 0.93 | 0.074 | 42.75 | 556.59 | 76.14 | 0.86 | 0.071 |
| **SCENEDIRECTOR (OURS)** | **38.29** | **516.83** | **76.85** | **0.81** | **0.052** | 35.83 | 464.56 | **77.02** | **0.78** | **0.052** |
| *Ours (Multi-View, Obj. Only)* | 35.18 | 305.11 | 76.78 | 0.90 | 0.055 | | | – | | |
| *Ours (Multi-View, Obj. + Traj.)* | 36.65 | 326.16 | 76.40 | 0.95 | 0.056 | | | – | | |

*DriveEditor* (Liang et al., 2025b), a driving-specific diffusion framework that leverages 3D layout projections and reference-image priors for controllable object manipulation. Since both baselines are limited to single-view, single-object editing, we apply them sequentially (deletion → replacement → repositioning → insertion) for Multi-Edit comparisons, whereas SceneDirector performs unified editing in a single inference pass. As our method inherently requires multi-view inputs, we align the evaluation by comparing our front-view results against the single-view baselines.

**Metrics.** We evaluate generation quality using FID (Heusel et al., 2017) and FVD (Unterthiner et al., 2018), and semantic alignment via CLIP-I (Huang et al., 2024b). To evaluate the downstream perception performance, we employ a PGD detector (Wang et al., 2022) pre-trained on WOD. We report detection Recall to assess the efficacy of the edits, along with Average Translation Error (ATE) and Average Orientation Error (AOE) to quantify geometric fidelity. To further assess whether the editing process preserves non-edited content, we report masked PSNR and masked LPIPS over background regions in Appendix B.1.

**Main Results.** Table 1 and Figure 4 present the comparison. **(1) Versatile Editing:** To ensure a fair comparison, we establish a Single-Edit protocol that aligns with the baselines' native capabilities. In *Single-Edit*, SceneDirector

achieves the best geometric alignment, reducing ATE/AOE to 0.78m/0.052. Despite its 14B-scale capacity, VACE-14B achieves only marginally better FID/FVD in this setting, yet suffers from substantially larger geometric error than ours (ATE 1.02m vs. 0.78m), with qualitative results showing misalignment and object hallucination. In *Multi-Edit*, sequentially applying single-edit baselines accumulates artifacts and control errors, whereas SceneDirector performs all edits jointly in one pass and achieves the best performance across all Multi-Edit metrics (FVD 516.83 vs. 729.42). **(2) Unified Capability:** We further analyze simultaneous object and trajectory editing under the 2m Gradual Transition protocol in the multi-view setting. Compared to the static-trajectory multi-view baseline (*Obj. Only*), the unified setting (*Obj. + Traj.*) introduces only marginal degradation (ATE 0.90m → 0.95m; AOE 0.055 → 0.056). These results show that SceneDirector can effectively perform object and ego-trajectory editing jointly in a single inference pass.

### 4.2. Ego-trajectory Editing

**Evaluation Benchmark.** To assess ego-trajectory controllability, we curate 64 evaluation scenarios from the WOD validation split via an automated pipeline (detailed in the Appendix C). Each scenario is evaluated under two deviation magnitudes (2m and 3m) and two trajectory modification

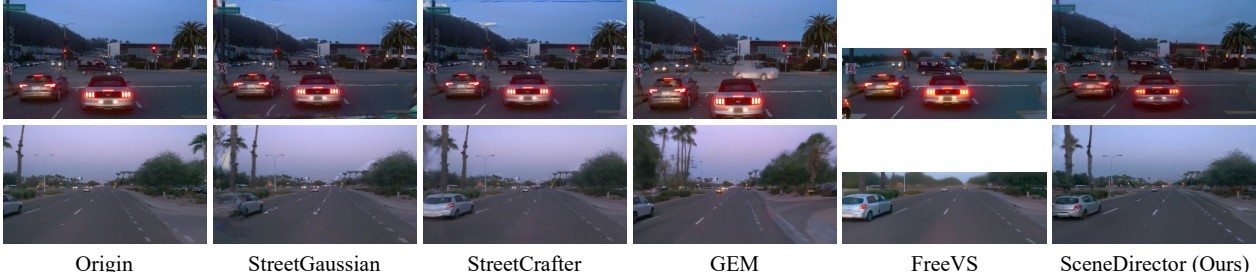

| Origin | StreetGaussian | StreetCrafter | GEM | FreeVS | SceneDirector (Ours) |

*Figure 5.* Qualitative comparison of trajectory editing under **Gradual Transition**. *StreetGaussian* fails to generalize to novel poses, showing severe distortions and artifacts. While *StreetCrafter* improves texture via diffusion (*StreetCrafter-DM*), it lacks explicit in-painting capabilities for disoccluded regions (e.g., behind trees), resulting in gray voids due to missing geometry. *GEM* displays weak control fidelity with noticeable structural loss. *FreeVS* produces incomplete frames by masking out the sky due to sparse LiDAR coverage. *SceneDirector* effectively hallucinates plausible details in geometric voids (e.g., sky and occluded background) while maintaining rigorous structural consistency, yielding artifact-free video synthesis. Additional results for **Fixed Offset** are provided in the Appendix E.

*Table 2.* Visual quality of trajectory editing under gradual transition and fixed offset settings. In the front-view setting, SceneDirector consistently outperforms all diffusion-based baselines across trajectory modes. SceneDirector+SG performs best under gradual-transition settings. SceneDirector-MV surpasses all baselines in the multi-view setting, showing robust visual fidelity under viewpoint changes.

| | | GRADUAL TRANSITION | | | | FIXED OFFSET | | | |
| | | 2M DEVIATION | | 3M DEVIATION | | 2M OFFSET | | 3M OFFSET | |
| METHOD | CATEGORY | FID ↓ | FVD ↓ | FID ↓ | FVD ↓ | FID ↓ | FVD ↓ | FID ↓ | FVD ↓ |
|---|---|---|---|---|---|---|---|---|---|
| FREEVS | | 79.25 | 1208.9 | 79.52 | 1255.5 | 81.34 | 1229.9 | 84.65 | 1214.2 |
| GEM | DIFFUSION | 42.90 | 667.9 | 44.02 | 661.2 | - | - | - | - |
| STREETCRAFTER-DM | | - | - | - | - | 46.25 | 694.0 | 52.22 | 757.7 |
| **SCENEDIRECTOR (OURS)** | | 34.48 | 476.1 | 37.21 | 513.5 | 36.38 | 477.3 | 46.96 | 564.9 |
| STREETGAUSSIAN | | 27.37 | 434.9 | 35.64 | 535.0 | 43.02 | 585.2 | 59.97 | 774.4 |
| STREETCRAFTER | RECONSTRUCTION | 26.36 | 418.2 | 32.44 | 468.4 | **33.32** | **417.7** | **43.27** | **539.9** |
| **SCENEDIRECTOR+SG (OURS)** | | **25.70** | **400.1** | **31.24** | **453.5** | 35.64 | 449.5 | 45.33 | 566.0 |
| FREEVS-MV | | 69.05 | 898.5 | 69.43 | 927.3 | 73.83 | 936.5 | 75.06 | 940.5 |
| STREETGAUSSIAN-MV | MULTIVIEW | 54.22 | 549.5 | 58.86 | 643.2 | 68.25 | 701.3 | 78.15 | 868.1 |
| **SCENEDIRECTOR-MV (OURS)** | | **44.41** | **445.5** | **49.10** | **469.2** | **51.34** | **474.1** | **57.05** | **544.6** |

*Table 3.* Geometric accuracy of front-view trajectory editing under gradual transition and fixed offset settings (X-ERR in meters). F1, recall, and X-ERR measure structural alignment after trajectory editing. SceneDirector improves F1/X-ERR over diffusion-based methods across all settings, while SceneDirector+SG achieves the best F1 among reconstruction methods, confirming high structural accuracy.

| | GRADUAL TRANSITION | | | | | | FIXED OFFSET | | | | | |
| | 2M DEVIATION | | | 3M DEVIATION | | | 2M OFFSET | | | 3M OFFSET | | |
| METHOD | F1 ↑ | R ↑ | X-ERR ↓ | F1 ↑ | R ↑ | X-ERR ↓ | F1 ↑ | R ↑ | X-ERR ↓ | F1 ↑ | R ↑ | X-ERR ↓ |
|---|---|---|---|---|---|---|---|---|---|---|---|---|
| FREEVS | 40.3 | 38.7 | 0.814 | 37.0 | 34.2 | 0.867 | 42.6 | 36.0 | 0.712 | 44.9 | **43.7** | 0.817 |
| GEM | 19.4 | 17.6 | 1.130 | 17.7 | 15.8 | 1.129 | - | - | - | - | - | - |
| STREETCRAFTER-DM | - | - | - | - | - | - | 40.8 | 33.7 | 0.860 | 35.5 | 27.9 | 0.902 |
| **SCENEDIRECTOR (OURS)** | 55.1 | 52.5 | 0.603 | 51.4 | 46.3 | 0.640 | 54.0 | 50.2 | 0.659 | 46.6 | 42.7 | 0.768 |
| STREETGAUSSIAN | 59.2 | 54.0 | 0.535 | 53.1 | 45.9 | **0.576** | 55.4 | 48.1 | **0.580** | 42.0 | 33.3 | **0.722** |
| STREETCRAFTER | 59.5 | 54.9 | 0.541 | 54.1 | 48.4 | 0.596 | 56.0 | 51.0 | 0.631 | 46.8 | 41.3 | 0.744 |
| **SCENEDIRECTOR+SG (OURS)** | **60.4** | **56.7** | **0.528** | **55.4** | **50.4** | 0.590 | **57.4** | **53.6** | 0.622 | **47.5** | 41.2 | 0.740 |

modes: *Gradual Transition*, which simulates a smooth departure from the original path as in a lane change, and *Fixed Offset*, which maintains a constant spatial displacement to test sustained viewpoint control. The 3m setting further stresses the model with stronger geometric shifts.

**Baselines.** We evaluate ego-trajectory editing against baselines from two categories. **(1) Reconstruction-based frameworks** include *StreetGaussian* (Yan et al., 2024), a dynamic 3DGS method for decomposed driving-scene re-

construction, and *StreetCrafter* (Yan et al., 2025), a hybrid framework that combines video diffusion priors with 3DGS reconstruction. To further evaluate multi-view consistency, we introduce *SceneDirector+SG*, where we train StreetGaussian on our generated videos under the standard StreetCrafter reconstruction setting. This setup serves two purposes: compared with StreetGaussian, it tests whether our generated videos provide reliable multi-view supervision for 3DGS reconstruction; compared with StreetCrafter,

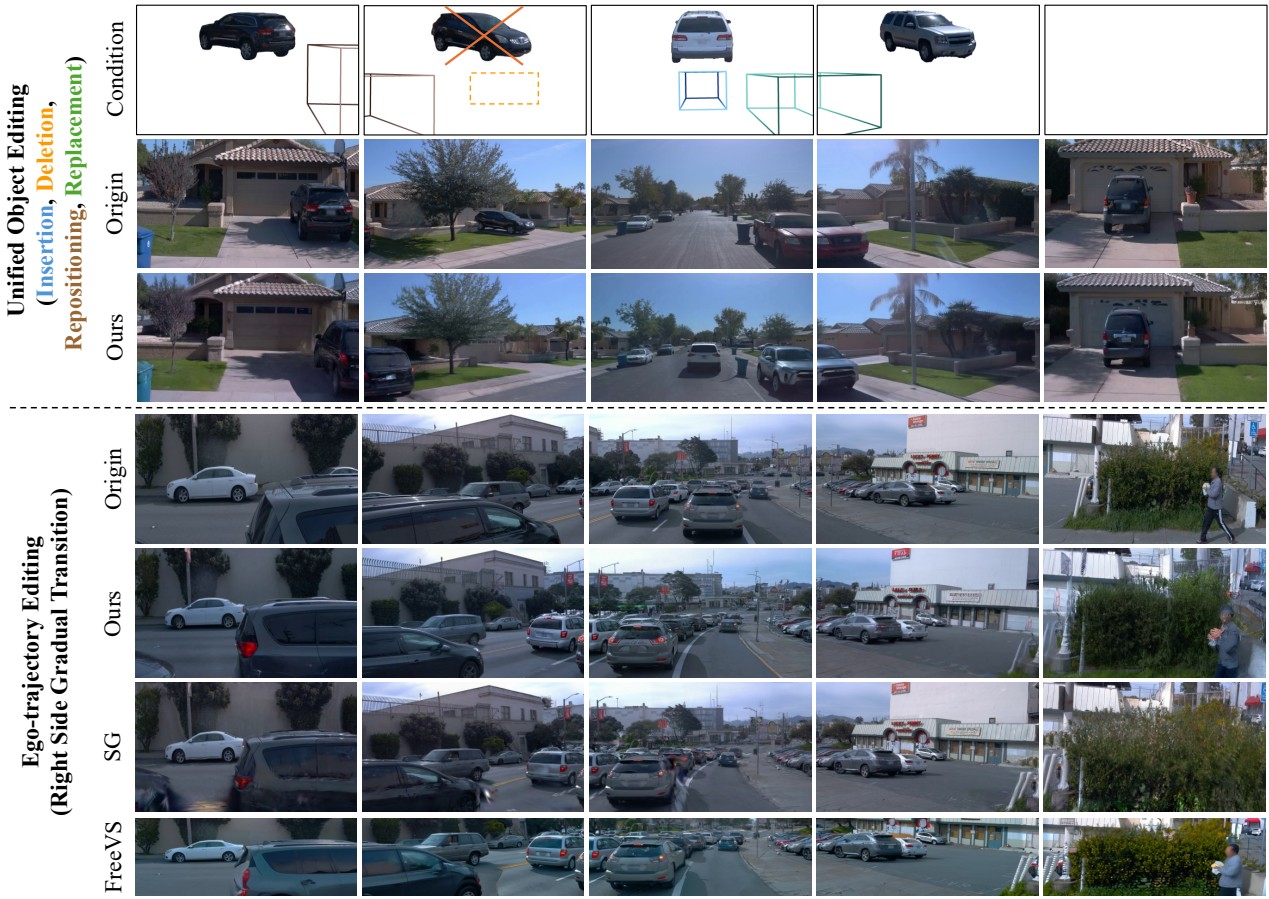

*Figure 6.* Qualitative results of SceneDirector on multi-view editing. **Top (Unified Object Editing):** Rows display the editing conditions (object assets and 3D bounding boxes), source frames, and our editing results. Our method enables unified editing in a single inference pass. **Bottom (Ego-trajectory Editing):** Results under a right-side gradual transition are shown for the source video, StreetGaussian (SG), FreeVS, and our method. SceneDirector better preserves cross-view structural consistency and photorealism than the baselines, which exhibit blur, artifacts, or incomplete regions under viewpoint changes, while faithfully following the specified editing controls.

it follows the same reconstruction-plus-diffusion paradigm, thereby isolating the quality of the generated diffusion priors. **(2) Diffusion-based methods** include *FreeVS* (Wang et al., 2025), which constructs LiDAR-based pseudo-view priors for trajectory-controlled view synthesis; *GEM* (Hassan et al., 2025), a trajectory-guided ego-vision world model driven by ego-motion and visual cues; and *StreetCrafter-DM*, the standalone diffusion component of StreetCrafter (Yan et al., 2025) without its 3DGS stage, which also uses LiDAR-rendered geometric conditions for camera-pose control. Following the control settings natively supported by each method, GEM is evaluated under Gradual Transition, while StreetCrafter-DM is evaluated under Fixed Offset.

**Metrics.** We assess performance in terms of generation quality and trajectory controllability. For visual quality, we likewise report FID and FVD. To quantify structural preservation under trajectory manipulation, we utilize a pre-trained 3D lane detector, Persformer (Chen et al., 2022), on the front-view videos. We treat lane detections from

the unedited source video as ground truth, which ensures a consistent relative benchmark because all methods are evaluated under identical deviation settings. For generated videos, the predicted lane points are projected back to the source coordinate system using the known ego poses to measure alignment. We report F1-score, Recall, and X-error (lateral error in meters) to evaluate the geometric accuracy.

**Main Results.** Tables 2, 3 and Figure 5 present quantitative and qualitative trajectory editing performance. SceneDirector consistently outperforms diffusion-based baselines in both visual quality and trajectory controllability. For example, under the front-view 2m Gradual Transition setting, it achieves much lower FVD than FreeVS and GEM (476.1 vs. 1208.9 and 667.9), while also improving structural alignment over FreeVS (F1 55.1 vs. 40.3). These results indicate that baselines struggle to maintain scene structure under camera-pose changes, whereas our geometric scaffold provides a stronger spatial anchor for novel-view synthesis. In Gradual Transition, SceneDirector+SG outper-

forms StreetCrafter in visual quality (FID 25.70 vs. 26.36 at 2m), and in Fixed Offset it maintains comparable visual quality while achieving better lane alignment (e.g., F1 47.5 vs. 46.8 at 3m). Since 3DGS reconstruction requires strict cross-view geometric consensus, these results suggest that SceneDirector produces videos that are temporally consistent and multi-view consistent, and can serve as effective priors for reconstruction, rather than only improving frame-level appearance. Figure 6 further provides multi-view qualitative comparisons. SceneDirector preserves cross-view geometry under unified editing, while avoiding the blurry artifacts of reconstruction-based methods and the structural discontinuities of diffusion-based baselines.

### 4.3. Ablation Study

We validate the components on the multi-view trajectory editing task (2m fixed offset and 3m gradual transition) using FID and FVD. We select this task because global view synthesis exposes both geometric errors and appearance artifacts, making it more discriminative than object editing.

**Impact of Reference Injection.** As shown in Table 4, removing the Mask-Gated Reference Attention (MGRA) (*w/o Ref. Attn*) leads to a clear performance drop, with FVD increasing by +25.8 under 3m Deviation. In this variant, the model mainly relies on the geometric scaffold, which provides reliable layout but lacks rich appearance details. The degradation confirms that reference appearance priors are essential for recovering realistic textures and compensating for information loss in sparse point clouds.

**Necessity of Uncertainty-Aware Gating.** As shown in Figure 7, replacing MGRA with standard cross-attention (*Standard Attn*) results in severe ghosting artifacts. This suggests that naively injecting reference features is insufficient. In contrast, MGRA regulates the reference injection according to both the uncertainty mask and the evolving feature content. Visualization of the gate maps further reveals a noise-dependent adaptive mechanism. At early timesteps, high noise levels dominate $h_l$, forcing the gate to rely primarily on $M$. As denoising resolves semantic structures, the gate becomes content-adaptive: it discerns semantic complexity and selectively injects texture details into rich areas (e.g., vehicles, buildings), rather than strictly adhering to geometric confidence.

**Robustness to Geometric Quality.** Finally, replacing the depth completion backbone with LRRU (Wang et al., 2023) (*Alt. Geo*) yields only marginal degradation in the metrics (FID $51.34 \to 51.87$ in 2m Fixed Offset). While the drop is more noticeable under the challenging 3m Deviation setting (FID $49.10 \to 50.04$), the model remains stable. This indicates that SceneDirector is not tightly coupled to a specific depth completion model, and can accommodate scaffolds of varying quality from different depth estimators.

*Table 4.* Ablation study on reference injection, uncertainty-aware gating, and geometric scaffold quality. **w/o Ref. Attn:** removing MGRA and disabling reference feature injection. **Standard Attn:** replacing MGRA with ungated cross-attention. **Alt. Geometry:** replacing the depth completion backbone with LRRU.

| METHOD | 3M DEVIATION | | 2M OFFSET | |
|---|---|---|---|---|
| | FID ↓ | FVD ↓ | FID ↓ | FVD ↓ |
| **SCENEDIRECTOR** | **49.10** | **469.2** | **51.34** | **474.1** |
| 1. W/O REF. ATTN | 53.02 | 495.0 | 55.46 | 512.5 |
| 2. STANDARD ATTN | 50.85 | 481.8 | 53.02 | 493.4 |
| 3. ALT. GEOMETRY | 50.04 | 477.7 | 51.87 | 478.0 |

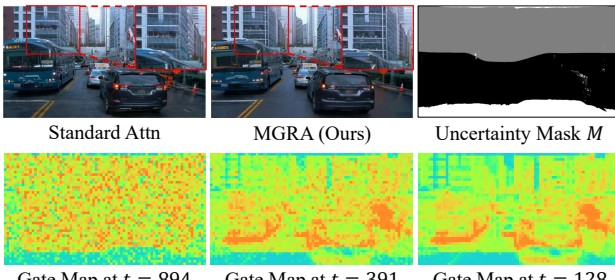

Standard Attn     MGRA (Ours)     Uncertainty Mask *M*

Gate Map at $t = 894$    Gate Map at $t = 391$    Gate Map at $t = 128$

*Figure 7.* Ablation study on Mask-Gated Reference Attention (MGRA). **Top:** MGRA eliminates ghosting artifacts observed in standard cross-attention. We visualize the mask $M$ (Black: sensor-verified; Gray: inferred layout; White: voids). **Bottom:** Evolution of gate values across diffusion timesteps $t$, where red indicates higher gate activation. The gating mechanism shifts from reliance on the mask prior under high noise ($t = 894$) to content adaptation ($t = 128$), selectively retrieving textures for semantic regions.

## 5. Conclusion

We propose SceneDirector, a unified framework for driving scene editing that bridges explicit geometry and generative priors. This reconciles the structural consistency required for trajectory control with the photorealistic synthesis essential for object manipulation. Key to this success is Mask-Gated Reference Attention, which leverages sensor uncertainty to harmonize the Unified Geometric Scaffold with texture priors from the Static Texture Bank. Extensive evaluations on the Waymo Open Dataset across object editing, trajectory editing, and multi-view settings demonstrate the superior controllability and quality of SceneDirector, offering a scalable solution for simultaneous object and ego-trajectory editing.

## 6. Limitations

While SceneDirector shows robust performance, we acknowledge certain limitations. First, SceneDirector uses LiDAR-guided depth completion to construct the scaffold. Although LiDAR is available in most driving datasets and is also adopted by several baselines, this requirement may limit camera-only applications. Second, although our model supports flexible camera setups, adapting it from the 5-camera Waymo setup to other layouts may require fine-tuning.

## Acknowledgements

This work made use of the HPC Platform of Huazhong University of Science and Technology.

## Impact Statement

This paper presents SceneDirector, a framework designed to advance the validation and robustness of autonomous driving systems by synthesizing diverse and safety-critical driving scenarios. By enabling the generation of rare "corner cases" without extensive physical testing, our work has the potential to improve road safety and reduce the environmental costs associated with real-world data collection.

However, we acknowledge that the generative capabilities of our method—specifically the ability to realistically alter vehicle trajectories and manipulate scene objects—carry potential risks. Like other high-fidelity video editing technologies, this framework could be misused to create misleading media or fabricate evidence (e.g., altered dashcam footage). Furthermore, relying on synthetic data for safety-critical applications requires rigorous validation to ensure that generated scenarios maintain physical realism and do not introduce biases that could compromise system reliability in the real world. We encourage the research community to develop robust detection mechanisms for synthetic media and to maintain strict oversight when utilizing generated data for safety validation.

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

# A. Network Architecture and Implementation Details

## A.1. Mask-Gated Reference Attention (MGRA)

The MGRA module dynamically modulates texture injection based on geometric uncertainty. The specific hyper-parameters are as follows:

- **Uncertainty Embedding ($\Psi$):** The discrete uncertainty mask $M$ is projected into a continuous embedding space with a dimension of 256.

- **Gating Network ($\mathcal{F}$):** The MLP takes the concatenation of the geometric feature and the mask embedding as input. It consists of a linear projection to a hidden dimension of 512, followed by a SiLU activation, and a final projection to a scalar output with Sigmoid activation.

- **Reference Attention:** We employ Multi-Head Cross-Attention with 16 heads. Each head has a dimension of 128 (2048/16).

- **Zero Initialization:** The MLP takes the coThe residual injection parameters of the reference branch are initialized to zero, ensuring that the model initially preserves the geometric structure.ncatenation of the geometric feature and the mask embedding as input.

## A.2. Training Configuration

Following the preparation strategy in Sec. 3.2, we generate 2364 clips for joint object-and-trajectory editing and retain 2394 original clips for trajectory-only editing. We initialize the model from the pre-trained Cosmos-Transfer2.5 weights and freeze the base blocks. The model is trained on 8 NVIDIA A800 GPUs for 40,000 iterations with a learning rate of $2 \times 10^{-5}$. We use a LambdaLinear scheduler with a linear warmup for the first $1,000$ steps. The effective global batch size is $4$. This is achieved using a data parallel size of $4$ with context parallel size 2. We use the FusedAdamW optimizer with weight decay $\lambda = 0.1$, and $\epsilon = 10^{-8}$. Training clips are processed at a resolution of $704 \times 1280$ with a sequence length of 29 frames.

# B. Additional Quantitative Evaluation

## B.1. Background Preservation in Object Editing

We further evaluate whether object editing preserves non-edited scene content. This evaluation is applied only to object editing, where the camera trajectory remains unchanged and the generated frames are spatially aligned with the original frames outside the edited regions. Ego-trajectory editing is excluded because viewpoint changes break pixel-level alignment, making PSNR and LPIPS unsuitable.

For each frame, we obtain an edited-region mask from the projected 3D regions of all editing operations, including insertion, deletion, replacement, and repositioning. The mask is slightly dilated to cover object boundaries and transition areas, and its complement is used as the background region. We compute masked PSNR only on these background pixels. For masked LPIPS, we replace the edited regions in the generated frame with the corresponding original pixels before computing LPIPS against the original frame. This protocol evaluates background fidelity without penalizing the intended object-level modifications.

As shown in Table 5, SceneDirector achieves the best background preservation in both single-object and multi-object editing. In the single-object setting, it improves PSNR over DriveEditor by $+0.9$ dB. The advantage is larger in multi-object editing, where sequential baselines accumulate background degradation across multiple editing steps, while SceneDirector performs all edits in a single inference pass.

## B.2. Cross-View Consistency Evaluation

To explicitly quantify the spatial coherence between adjacent camera views, we conduct a direct feature-level comparison in overlapping regions. While the reconstruction metrics in the main paper implicitly validate geometric consensus, this section provides a direct measurement of visual and semantic alignment across sensor boundaries.

*Table 5.* Background Preservation in Object Editing. PSNR ($\uparrow$) and LPIPS ($\downarrow$) are computed only over non-edited background regions. *SceneDirector* better preserves unrelated scene content under both single-object and multi-object editing.

| SETTING | METHOD | LPIPS$_{mask}$ $\downarrow$ | PSNR$_{mask}$ $\uparrow$ |
|---|---|---|---|
| SINGLE-OBJECT EDITING | **SCENEDIRECTOR** | **0.329** | **29.1** |
| | DRIVEEDITOR | 0.348 | 28.2 |
| | VACE | 0.360 | 27.8 |
| MULTI-OBJECT EDITING | **SCENEDIRECTOR** | **0.335** | **28.8** |
| | DRIVEEDITOR | 0.376 | 26.9 |
| | VACE | 0.374 | 27.3 |

### B.2.1. EVALUATION METRICS AND SETUP

**Overlap Definition and Camera Setup.** The Waymo Open Dataset utilizes a surround-view system consisting of five cameras: Side Left (SL), Front Left (FL), Front (F), Front Right (FR), and Side Right (SR). We evaluate consistency across the four adjacent overlapping interfaces: **SL-FL**, **FL-F**, **F-FR**, and **FR-SR**. For each adjacent pair, we crop the overlapping regions located at their shared boundaries (rightmost 1/3 of the left view and leftmost 1/3 of the right view).

**Metrics.** We employ two distinct feature extractors:

- **CLIP-I (Semantic Consistency):** Measures the cosine similarity of high-level semantic embeddings. High scores indicate consistent object identities across views.

- **DINOv2 (Structural Consistency):** Measures the cosine similarity of fine-grained geometric features. High scores indicate precise structural alignment without "drifting".

### B.2.2. QUANTITATIVE RESULTS

The evaluation results are presented in Table 6 (Semantic) and Table 7 (Structural).

For **Object Editing**, SceneDirector achieves a CLIP-I average of 0.850, close to the *Original Data* baseline of 0.858, indicating that the edited objects remain semantically consistent across views. It also achieves a DINOv2 average of 0.774, comparable to the original data reference of 0.769, showing that object editing does not disrupt cross-view structural coherence.

For **Trajectory Editing**, we observe distinct behaviors across scenarios:

- **Semantic Stability (Table 6):** SceneDirector achieves the best average CLIP-I scores in both *Gradual Transition* settings, outperforming FreeVS-MV and StreetGaussian-MV. In *Fixed Offset* settings, reconstruction-based StreetGaussian-MV remains slightly stronger due to its explicit representation, while SceneDirector stays competitive and consistently outperforms FreeVS-MV, suggesting that the synthesized visual concepts remain semantically stable under view shifts.

- **Structural Integrity (Table 7):** The diffusion-based baseline FreeVS-MV shows clear structural degradation in the challenging *Fixed 2m Offset* setting, where the average DINOv2 score drops to 0.683. In contrast, SceneDirector achieves a substantially higher score of 0.789, close to the *Original Data* reference of 0.798 and higher than StreetGaussian-MV's 0.779. Similar robustness is observed under *Fixed 3m Offset*, where SceneDirector obtains the best average score among generated results. These results support the effectiveness of the Unified Geometric Scaffold in preserving cross-view structural consistency under viewpoint shifts.

## C. Automated Evaluation Benchmark Construction

### C.1. Scenario Curation and Pre-processing

We applied our automated generation pipeline to the validation split. Since not every driving scene physically accommodates all editing types (e.g., due to specific road topology or traffic congestion), the pipeline first filtered for feasible operations. From the pool of successfully generated samples, we curated 64 representative scenarios for object editing and 64 scenarios

*Table 6.* Semantic Consistency Evaluation (CLIP-I). We measure the cosine similarity (↑) of CLIP features in overlapping regions. The "Original Data" serves as the real-world reference. *SceneDirector* achieves the best average scores in Gradual Transition scenarios. In Fixed Offset settings, it remains competitive with reconstruction-based StreetGaussian and outperforms the diffusion-based FreeVS, demonstrating robust semantic stability under view shifts.

| TASK | SCENARIO | METHOD | SL-FL | FL-F | F-FR | FR-SR | AVG. |
|---|---|---|---|---|---|---|---|
| OBJECT EDITING | - | ORIGINAL DATA | 0.847 | 0.866 | 0.867 | 0.853 | 0.858 |
| | | **SCENEDIRECTOR** | **0.838** | **0.861** | **0.863** | **0.839** | **0.850** |
| | REAL DATA | ORIGINAL DATA | 0.863 | 0.878 | 0.874 | 0.859 | 0.868 |
| | GRADUAL 2M DEV. | FREEVS-MV | 0.857 | 0.881 | 0.875 | 0.844 | 0.864 |
| | | STREETGAUSSIAN-MV | **0.874** | 0.874 | 0.869 | **0.876** | 0.873 |
| | | SCENEDIRECTOR | 0.863 | **0.889** | **0.887** | 0.873 | **0.878** |
| TRAJECTORY EDITING | GRADUAL 3M DEV. | FREEVS-MV | 0.854 | 0.882 | 0.872 | 0.841 | 0.862 |
| | | STREETGAUSSIAN-MV | **0.872** | 0.873 | 0.866 | **0.876** | 0.872 |
| | | SCENEDIRECTOR | 0.863 | **0.889** | **0.886** | 0.875 | **0.878** |
| | FIXED 2M OFF. | FREEVS-MV | 0.774 | 0.784 | 0.782 | 0.794 | 0.783 |
| | | STREETGAUSSIAN-MV | **0.861** | **0.884** | **0.886** | **0.873** | **0.876** |
| | | SCENEDIRECTOR | 0.849 | 0.874 | 0.869 | 0.837 | 0.857 |
| | FIXED 3M OFF. | FREEVS-MV | 0.848 | 0.871 | 0.868 | 0.837 | 0.856 |
| | | STREETGAUSSIAN-MV | 0.865 | **0.885** | **0.887** | 0.872 | **0.877** |
| | | SCENEDIRECTOR | **0.869** | 0.870 | 0.869 | **0.874** | 0.870 |

*Table 7.* Structural Consistency Evaluation (DINOv2). We measure the cosine similarity (↑) of DINOv2 features to assess geometric alignment. The "Original Data" indicates the inherent cross-view coherence of real-world videos. *SceneDirector* maintains strong structural consistency across most settings and is especially robust in Fixed Offset scenarios, where FreeVS degrades substantially, e.g., 0.683 avg in Fixed 2m versus 0.789 avg for our method.

| TASK | SCENARIO | METHOD | SL-FL | FL-F | F-FR | FR-SR | AVG. |
|---|---|---|---|---|---|---|---|
| OBJECT EDITING | - | ORIGINAL DATA | 0.729 | 0.806 | 0.806 | 0.735 | 0.769 |
| | | **SCENEDIRECTOR** | **0.727** | **0.819** | **0.822** | **0.727** | **0.774** |
| | REAL DATA | ORIGINAL DATA | 0.765 | 0.834 | 0.829 | 0.763 | 0.798 |
| | GRADUAL 2M DEV. | FREEVS-MV | 0.722 | **0.842** | 0.831 | 0.704 | 0.775 |
| | | STREETGAUSSIAN-MV | 0.746 | 0.832 | 0.821 | **0.731** | 0.783 |
| | | SCENEDIRECTOR | **0.768** | **0.842** | **0.842** | 0.725 | **0.794** |
| TRAJECTORY EDITING | GRADUAL 3M DEV. | FREEVS-MV | **0.821** | 0.843 | 0.831 | **0.802** | **0.824** |
| | | STREETGAUSSIAN-MV | 0.744 | 0.831 | 0.821 | 0.727 | 0.781 |
| | | SCENEDIRECTOR | 0.767 | **0.844** | **0.842** | 0.722 | 0.794 |
| | FIXED 2M OFF. | FREEVS-MV | 0.664 | 0.706 | 0.703 | 0.659 | 0.683 |
| | | STREETGAUSSIAN-MV | 0.741 | 0.824 | 0.821 | **0.728** | 0.779 |
| | | SCENEDIRECTOR | **0.762** | **0.839** | **0.838** | 0.717 | **0.789** |
| | FIXED 3M OFF. | FREEVS-MV | 0.716 | **0.842** | 0.833 | 0.699 | 0.772 |
| | | STREETGAUSSIAN-MV | 0.737 | 0.823 | 0.821 | **0.724** | 0.776 |
| | | SCENEDIRECTOR | **0.762** | 0.833 | **0.837** | 0.717 | **0.787** |

for trajectory editing. These selected scenarios cover diverse driving conditions, including varying traffic densities, complex turns, and straight roads, ensuring a balanced evaluation.

## C.2. Object Editing Pipeline

For each processed clip, the pipeline identifies valid manipulation targets based on temporal stability (tracked for $\geq 16$ frames) and semantic class (cars/trucks).

**1. Object Insertion.** Insertion requires placing new objects that adhere to traffic rules and physical constraints. We employ a multi-stage heuristic planner:

- **Candidate Generation:** We generate candidate trajectories relative to a *Reference Anchor* (the dominant traffic flow or ego-vehicle) using three strategies in descending priority:

1. *Longitudinal Gap Filling:* The pipeline scans for longitudinal gaps between the reference vehicle and its neighbors. If a gap exceeds the safety threshold ($L_{obj} + 6.0m$), the candidate is placed at the gap's midpoint.
2. *Platoon Formation:* Candidates are placed at fixed longitudinal offsets (e.g., $\pm 10m$) to simulate car-following.
3. *Adjacent Lane Injection:* Candidates are spawned in adjacent lanes with lateral offsets of $\pm 3.6m$.

- **Ground Alignment:** We estimate the local ground elevation $z_{ground}(t)$ by interpolating the vertical positions of neighboring vehicles to prevent floating artifacts.

- **Physics and Visibility Validation:** Candidates are accepted only if they pass collision avoidance (OBB intersection test) and frustum visibility checks (visible pixel area > 200 pixels).

**2. Object Repositioning.** Objects are shifted along their local axes with a fixed magnitude of $\Delta d = 3.0m$. The new position undergoes strict collision checks against static background geometry and dynamic agents.

**3. Object Deletion and Replacement.** The algorithm automatically identifies prominent foreground objects for **Deletion** (removing rendering instructions) or **Replacement** (swapping 3D asset IDs while maintaining original trajectories), creating pairs for inpainting and semantic generation evaluation.

### C.3. Ego-trajectory Editing Pipeline

For ego-trajectory editing, we synthesize novel view sequences under two modes: *Fixed Offset* and *Gradual Transition*. To ensure physical realism, we implement a strict feasibility screening and a kinematic-aware trajectory parameterization.

**1. Feasibility Screening via Occupancy Analysis.** Before generation, we define a "virtual ego-vehicle" using the data collection vehicle's dimensions. A proposed deviation is deemed valid only if it satisfies two constraints:

- **Dynamic Collision Check:** We perform an Oriented Bounding Box (OBB) intersection test between the virtual ego-vehicle and all labeled dynamic objects at every timestamp.

- **Drivable Region Constraint:** Utilizing the HD Map, we identify free space by rasterizing lane boundaries. We discard trajectories where the vehicle footprint encroaches on non-drivable areas (e.g., sidewalks) or crosses solid lane lines into opposing traffic.

**2. Trajectory Parameterization (Gradual Transition).** To simulate realistic lane-change maneuvers, we model the trajectory by simultaneously modulating the lateral shift and the vehicle's heading (yaw). Let $T$ be the maneuver duration and $S$ be the total lateral shift. We define a normalized time $\tau = \text{clip}(t/T, 0, 1)$.

- **Lateral Displacement (Half-Cosine Ease):** To ensure smooth acceleration and deceleration, the lateral offset $\Delta y_t$ follows a half-cosine curve:

$$\Delta y_t = \frac{S}{2} \cdot (1 - \cos(\pi\tau)) \tag{7}$$

  This curve ensures zero lateral velocity at both the start and end of the maneuver.

- **Yaw Adaptation (Sine Wave):** Simply translating the camera laterally introduces unnatural sliding artifacts. To mimic realistic steering (turning into the lane and straightening out), we introduce a yaw adjustment $\Delta\psi_t$ modeled by a sine wave:

$$\Delta\psi_t = \psi_{peak} \cdot \sin(\pi\tau) \tag{8}$$

  where $\psi_{\text{peak}} \approx \min(5°, 2°|S|)$, with $S$ measured in meters. This rotates the view towards the target lane during the shift and returns to the original heading ($\Delta\psi = 0$) upon completion.

- **Pose Synthesis:** The final target pose $P'_t$ is computed by transforming the original pose $P_t$ with the local lateral translation $T_{lat}$ and rotation $R_z$:

$$P'_t = P_t \cdot T_{lat}(\Delta y_t) \cdot R_z(\Delta\psi_t) \tag{9}$$

## D. Self-Supervised Training Data Generation

To enable self-supervised training without paired ground truth, we construct synthetic training triplets $(\mathcal{S}, \mathcal{C}, V_{gt})$ using two procedures: 3D asset curation for geometric scaffolding and synthetic view perturbation for trajectory misalignment.

### D.1. 3D Asset Curation Pipeline

To construct the 3D asset library $\mathcal{A}_{obj}$ used in the Unified Geometric Scaffold, we adopt a three-stage pipeline: visibility filtering, instance segmentation, and semantic verification.

**1. Automated Visibility Filtering.** We first parse the raw driving logs to identify potential object candidates. To ensure sufficient multi-view coverage and visibility, we filter objects based on their projection on the camera image plane:

- **Overlap Threshold:** We calculate the intersection between the projected 3D bounding box and the camera canvas. An object is considered visible in a frame only if the intersection ratio exceeds a threshold $\tau_{overlap} = 0.35$.

- **Temporal Stability:** To guarantee robust reconstruction, a candidate must remain visible for at least one-third of the frames within the target video chunk, filtering out transient observations.

**2. Instance Segmentation and Pre-processing.** For candidates passing the visibility filter, we employ SAM 2 (Ravi et al., 2025) to extract pixel-perfect masks. We utilize the projected 2D bounding boxes as box prompts for the SAM 2 image predictor. We crop the target object and remove small disconnected noisy regions with an area ratio below 0.06. Candidates with a spatial resolution below 120 pixels are discarded to ensure sufficient textural detail.

**3. VLM-Based Semantic Verification.** Automated segmentation may occasionally yield artifacts, such as truncated vehicles or mis-segmented background elements. We deploy Qwen2.5-VL-7B (Bai et al., 2025) as a semantic verifier to conduct a visual quality inspection.

- **Heuristic Edge Detection:** Before VLM inference, we apply a contour-based heuristic to detect straight edges aligned with image borders. Objects with significant edge truncation are rejected early to save computational costs.

- **Structured Verification:** We design a structured prompt that asks the VLM to inspect each candidate and return a JSON-formatted decision. The model evaluates each asset against four strict criteria:
  1. **Content Check:** Verifying the subject is a vehicle (car, truck, bus) and not background noise.
  2. **Completeness Check:** Ensuring core components (body, roof, wheels) are not truncated or occluded.
  3. **Quality Check:** Rejecting images with severe motion blur or low-light degradation.
  4. **Segmentation Precision:** Checking for jagged edges or the inclusion of detached background elements (e.g., ground patches).

### D.2. Synthetic View Perturbation Function

The perturbation function $\Phi$ transforms the ground truth video $V_{gt}$ into a structurally misaligned reference $V_{ref}$ to force the model to rely on the geometric scaffold for alignment. This function $\Phi$ simulates trajectory deviations via decoupled temporal and spatial warping strategies, with kinematic smoothing and canvas filling constraints to approximate realistic ego-motion while preventing shortcut learning from invalid borders.

**Temporal Resampling.** To simulate longitudinal misalignment (e.g., velocity differences), we apply a speed scaling factor. We resample the video frames via temporal interpolation, effectively compressing or expanding the timeline to mimic acceleration or deceleration relative to the original log.

**Spatial Warping.** To simulate lateral deviations (e.g., lane changes), we apply view-dependent 2D image warping. The perturbation follows a kinematic trajectory defined by a lateral shift $\Delta x(t)$ and a yaw adjustment $\Delta \psi(t)$.

- **Kinematic Smoothing:** To produce smooth motion, the lateral shift follows a *half-cosine ease curve* with zero velocity at the start and end. Simultaneously, the yaw angle follows a *sine wave* peaking at $\sim 5°$ to mimic steering-induced heading changes during lane-change maneuvers.

- **View-Dependent Transforms:** We approximate 3D parallax effects using different 2D warping strategies based on the camera view index:
  1. *Front and Diagonal Views (Affine):* For forward-facing cameras, we apply an Affine transformation combining translation, rotation, and shear. The shear component is crucial to approximate the shift in vanishing points during lateral movement.



StreetGaussian StreetCrafter FreeVS SceneDirector (Ours)

*Figure 8.* **Qualitative comparison of geometric consistency via lane detection.** We visualize the 3D lane predictions (shown in the 3D grid view on the left of each pair) and their reprojection onto the generated frames (shown on the right).

    2. *Side Views (Perspective):* For side-facing cameras, affine transforms fail to capture the depth-dependent parallax. We explicitly compute a Perspective transformation (Homography) by warping the image corners, simulating the non-linear distortion of objects passing the field of view.

- **Canvas Filling Constraint:** Warping operations (rotation/shear) typically introduce invalid black borders. To prevent the network from learning trivial shortcuts from these artifacts, we solve for a minimal scaling factor $s_{min} \geq 1.0$ via binary search. This ensures that the warped image covers the target canvas and avoids invalid borders.

## E. Additional Visualization Results

In this section, we provide additional qualitative results on structural integrity and photorealism.

### E.1. Geometric Consistency via Lane Detection

To explicitly verify whether the synthesized driving scenes preserve valid road topology suitable for downstream perception tasks, we employ a state-of-the-art 3D lane detector, Persformer (Chen et al., 2022), to analyze the generated videos.

Figure 8 presents the visualization of detected 3D lanes and their 2D projections across different methods.

### E.2. Geometric Precision via Object Detection

Beyond lane topology, we further assess the precision of object manipulation by running a pre-trained 3D object detector on the edited videos. This evaluation serves two purposes: (1) verifying that inserted/modified objects are realistically rendered such that perception algorithms can recognize them, and (2) quantifying the alignment between the user-specified target position (Ground Truth) and the actual generated content. Figure 9 visualizes the detection results across different editing tasks. As observed:

- **Baselines:** *VACE* and *DriveEditor* struggle to strictly adhere to the geometric instructions. We observe significant **geometric drift**, where the detected boxes (filled) deviate noticeably from the target layout (wireframe). Furthermore, they exhibit a high rate of **false positives** (hallucinations), recognizing background artifacts as targets.

- **Ours:** *SceneDirector* demonstrates strong geometric fidelity. The detection results closely align with the target boxes, indicating that our method places objects according to the specified 3D constraints. Moreover, the cleaner background reduces false positives, further supporting the quality of the generated results.

### E.3. Qualitative Results on Fixed Offset Trajectory Editing

Complementing the "Gradual Transition" analysis in the main text (Figure 5), we present comparative results for the **Fixed Offset** setting in Figure 10. This task requires maintaining a constant lateral deviation throughout the sequence, demanding continuous hallucination of occluded regions. As shown in the visualization:

- **Baselines:** Reconstruction-based methods (e.g., *StreetGaussian*, *StreetCrafter*) exhibit characteristic rendering artifacts, such as blurring or streaking in novel views.

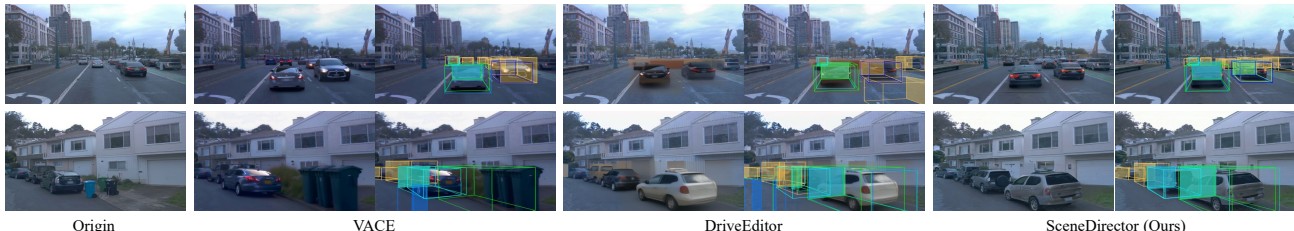

| Origin | VACE | DriveEditor | SceneDirector (Ours) |

*Figure 9.* **Qualitative evaluation of editing precision via 3D object detection.** The leftmost column displays the original scene, followed by the editing results from *VACE*, *DriveEditor*, and *SceneDirector*. **Top Row:** A complex combination of object insertion and repositioning (movement). **Bottom Row:** Simultaneous object insertion and replacement. **Legend:** Filled semi-transparent boxes indicate the detection results from the perception model, while hollow wireframe boxes represent the target Ground Truth (GT) layout. **Analysis:** Baseline methods exhibit hallucinations, including false positive boxes in empty areas, and geometric misalignment, where detection boxes drift away from the target layout. In contrast, *SceneDirector* achieves closer alignment between detections and GT constraints with fewer artifacts, indicating its potential for generating perception-oriented evaluation data.

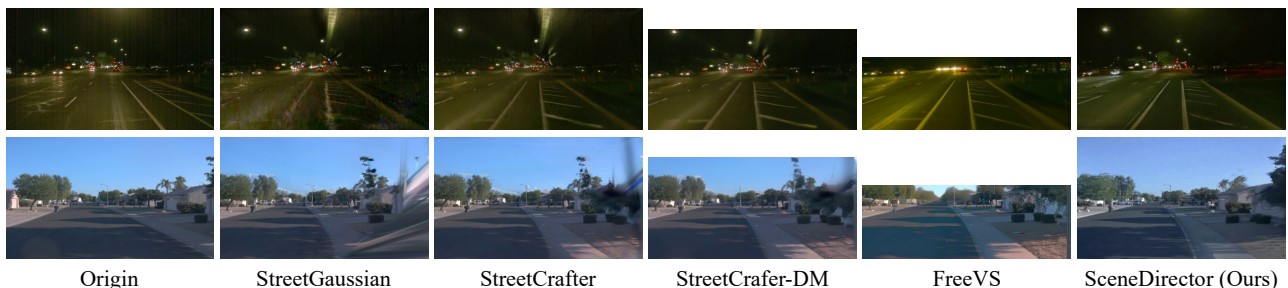

| Origin | StreetGaussian | StreetCrafter | StreetCrafer-DM | FreeVS | SceneDirector (Ours) |

*Figure 10.* **Qualitative comparison of Fixed Offset trajectory editing.** We compare methods under a constant lateral shift trajectory in both night (top) and day (bottom) scenarios. **Baselines:** *StreetGaussian* and *StreetCrafter* show noticeable rendering distortions. **Ours:** *SceneDirector* synthesizes full-frame, photorealistic videos with precise structural alignment and plausible details in occluded regions.

*Table 8.* **Time cost comparison for generating a 29-frame video clip.** "Scope" indicates the spatial and objective coverage of a single inference pass. *SceneDirector* processes multiple objects across surround views at $1280 \times 704$ in ∼15 minutes, while several baselines operate on a single view, a single object, or lower resolutions.

| TASK | METHOD | # GPUs | RESOLUTION | SCOPE | TIME (MIN) |
|---|---|---|---|---|---|
| OBJECT EDITING | VACE | 2 | $830 \times 480$ | 1 VIEW & 1 OBJECT | ∼ 8 |
| | DRIVEEDITOR | 1 | $1024 \times 576$ | 1 VIEW & 1 OBJECT | ∼ 5 |
| | **SCENEDIRECTOR (OURS)** | **2** | **$1280 \times 704$** | **5 VIEWS & 4 OBJECTS** | **∼ 15** |
| TRAJECTORY EDITING | GEM | 1 | $1024 \times 576$ | 1 VIEW | ∼ 3 |
| | FREEVS | 1 | $660 \times 380$ | 5 VIEWS | ∼ 6 |
| | **SCENEDIRECTOR (OURS)** | **2** | **$1280 \times 704$** | **5 VIEWS** | **∼ 15** |
| | STREETCRAFTER | 1 | $1600 \times 1072$ | 1 VIEW | ∼ 60 |
| | STREETGAUSSIAN | 1 | $1600 \times 1072$ | 5 VIEWS | ∼ 40 |
| | **SCENEDIRECTOR+SG (OURS)** | **2** | **$1600 \times 1072$** | **5 VIEWS** | **∼ 55** |

- **Ours:** *SceneDirector* generates more complete and photorealistic frames with better preserved geometry and consistent lighting across both night and day scenarios.

# F. Efficiency Analysis

We assess inference efficiency on NVIDIA A800 GPUs. Note that SceneDirector and VACE-14B utilize 2 GPUs due to memory requirements, while other baselines operate on a single GPU. To ensure a fair comparison, all diffusion-based methods are evaluated using the default sampling steps specified in their respective original implementations.

**Object Editing.** As shown in Table 8, baselines such as VACE and DriveEditor are restricted to processing a single object within a single view per inference pass. For instance, VACE ($830 \times 480$) requires ∼8 minutes for this limited scope. In contrast, SceneDirector performs unified editing for **4 objects across 5 views** at a higher resolution ($1280 \times 704$) in a

comparable duration ($\sim$15 minutes). Despite the heavier workload and higher resolution, our parallelized architecture improves per-pass coverage by avoiding iterative single-object or single-view inference.

**Trajectory Editing.** Reconstruction-based methods (e.g., StreetGaussian, StreetCrafter) rely on per-scene optimization at high resolutions ($1600 \times 1072$), leading to variable and lengthy training times. SceneDirector does not require per-scene optimization and generates novel trajectories directly by inference at $1280 \times 704$.

