# OpenReview forum: "SceneDirector: Bridging Explicit Geometry and Generative Priors for Unified Driving Scene Editing"
_ICML.cc/2026/Conference — ICML 2026 regular_

### Official Review · Reviewer_DF23 · 2026-03-06

**Soundness:** 2
**Presentation:** 3
**Significance:** 3
**Originality:** 3
**Overall Recommendation:** 5
**Confidence:** 3

**Summary:**

This paper introduces "SceneDirector," a diffusion-based framework designed to bridge explicit geometry and generative priors for unified driving scene editing. Addressing the challenge of simultaneously editing objects and ego-trajectories, the authors propose a Unified Geometric Scaffold constructed via LiDAR-guided depth completion to provide structural guidance. To leverage generative priors, the source video is encoded into a Static Texture Bank. A key contribution is the Mask-Gated Reference Attention mechanism, which dynamically regulates the interaction between the geometric scaffold and the texture bank based on geometric uncertainty. Extensive evaluations demonstrate that SceneDirector outperforms state-of-the-art methods in terms of both controllability and visual quality.

**Compliance With Llm Reviewing Policy:**

Affirmed.

**Final Justification:**

Overall, the authors have made solid contributions and successfully addressed my concerns during rebuttal.

**Key Questions For Authors:**

Overall, I do not have major concerns regarding this work. The paper presents a novel and well-motivated approach to unified driving scene editing with impressive results. The weaknesses noted are primarily requests for further clarification and more comprehensive metric reporting. I consider this to be a good paper that makes a solid contribution to the field.

**Limitations:**

Yes.

**Strengths And Weaknesses:**

**Strengths**

1. The framework is strategically built upon the timely and advanced video world model, Cosmos-Transfer. Furthermore, the authors ensure a rigorous evaluation by comparing their method against the most recent and relevant baselines in the field, ensuring the proposed approach is benchmarked against the current state-of-the-art.

2. The methodology is articulated with great clarity, making the technical approach and architectural choices easy to understand. The experimental results provide compelling evidence of the method's effectiveness, showing obvious qualitative and quantitative improvements over existing baselines.

3. The motivation for addressing the simultaneous editing of 3D box-defined objects and ego-trajectories is well-defined. This dual capability addresses a significant gap in generating diverse and realistic validation scenarios for autonomous driving systems, making the problem definition highly relevant.

**Weaknesses**

1. Regarding the proposed self-supervised pair construction, it remains unclear how accurately this method can represent the ground truth video corresponding to novel trajectories or modified objects. Specifically, for the Synthetic View Perturbation Function, the authors should clarify if this is a mainstream function used to simulate misalignment in this domain. If it is, relevant references should be cited; if not, a deeper reasoning and the principles behind this specific design choice need to be provided.

2. The proposed method explicitly depends on LiDAR data for depth completion to construct the scene geometry. However, some of the compared baselines, such as DriveEditor, do not always use or assume the availability of LiDAR data. This discrepancy in sensor prerequisites makes the comparison slightly unbalanced, and the authors should discuss the implications of this dependency on hardware availability.

3. Some standard evaluation metrics appear to be missing. For tasks involving object editing or novel view synthesis, it is crucial to evaluate the similarity of the background scene before and after editing to ensure consistency. Metrics such as PSNR and LPIPS should be reported to measure pixel-level fidelity and perceptual differences, ensuring a more comprehensive and standard assessment of the proposed method's performance.

---

> ### Author Rebuttal · Authors · 2026-03-31
>
> We sincerely thank you for recommending acceptance and recognizing our work as *"a good paper that makes a solid contribution to the field."* We are encouraged that you found our methodology *"articulated with great clarity,"* and the motivation *"well-defined"* and *"highly relevant."*
>
> Such encouraging feedback reflects a broader consensus among the reviewers, who jointly emphasize:
> 1. The practical importance of unifying object and trajectory editing.
> 2. The principled and coherent design of our multi-component pipeline.
> 3. The strength and validity of our experimental results.
>
> We address your remaining concerns in detail below.
>
> ---
> ## Q1: Clarification on Synthetic View Perturbation function.
> We appreciate the opportunity to clarify this. As paired real-world data for driving scene editing does not exist, **we follow an established paradigm in this field** of artificially degrading ground-truth videos to simulate inference-time discrepancies. The relevant methods and the specific view perturbation techniques they adopt are summarized below:
> * **Monodepth2** *utilizes spatial warping and temporal frame interpolation* to construct self-supervised pairs for ego-motion estimation.
> * **FreeVS** *employs Viewpoint Transformation Simulation*, sampling temporally mismatched frames to construct pseudo-images that simulate camera movement.
> * **FreeSim** *uses Gaussian perturbation and extrapolated rendering* to construct degraded training pairs.
>
> We **refine this paradigm for our unified editing** via: **(1) Kinematic smoothing** (half-cosine lateral shift and sinusoidal yaw) to match real lane-change dynamics; **(2) View-dependent warping** (affine for front cameras, perspective homography for side cameras) to approximate 3D parallax; and **(3) Canvas filling constraints** to prevent shortcut learning from black borders.
>
> The effectiveness of this paradigm is validated by strong generalization on the WOD validation split: SceneDirector achieves SOTA FID (34.48) and FVD (476.1) for trajectory editing (Table 2), and the lowest ATE/AOE for object editing (Table 1). We will cite these foundational works and strengthen this discussion in the revised paper.
>
> ---
> ## Q2: LiDAR dependency and comparison fairness.
> We thank you for pointing this out. Indeed, we acknowledge our reliance on LiDAR depth completion as a limitation in Appendix G. However, this design choice is justified and equitable:
>
> **LiDAR is a standard modality in autonomous driving.** Major datasets (WOD, nuScenes, Argoverse 2, and ONCE) and Level 4 platforms (Waymo and Cruise) universally equip LiDAR, aligning our framework with real-world deployments.
>
> **Spatially consistent editing requires explicit 3D geometry.** The baseline *FreeVS also relies on LiDAR* to construct pseudo-view priors. Other baselines like StreetCrafter demand expensive, time-consuming per-scene optimization, utilizing 3DGS to explicitly reconstruct the 3D scene.
>
> **Our approach, by leveraging LiDAR, unlocks highly efficient, unified scene control.** Fusing LiDAR depth with 3D assets constructs a training-free Unified Geometric Scaffold. This explicit spatial anchor directly enables simultaneous object and ego-trajectory editing in a single inference pass, bypassing massive per-scene optimization overhead. We believe this paradigm paves the way for future scalable autonomous driving simulations.
>
> We will expand on these sensor prerequisites in the revision.
>
> ---
> ## Q3: Missing PSNR and LPIPS metrics.
> We thank you for this valuable suggestion.
> 1. **Trajectory Editing:** PSNR and LPIPS require strict spatial pixel alignment, making them mathematically unsuitable for novel view synthesis where camera viewpoints shift. Instead, alignment-free feature similarities like CLIP-I and DINOv2 (see Appendix B) can effectively reflect background and structural preservation.
> 2. **Object Editing:** For static-viewpoint object editing, we agree these metrics are highly valuable. We computed PSNR and LPIPS exclusively on the non-edited background regions to evaluate pixel-level fidelity:
>
> | Setting | Method | LPIPS$_\text{masked}$ $\downarrow$ | PSNR$_\text{masked}$ $\uparrow$ |
> |-|-|:-:|:-:|
> | **Single-Object Editing** | SceneDirector (Ours) | **0.329** | **29.1** |
> | | DriveEditor | 0.348 | 28.2 |
> | | VACE | 0.360 | 27.8 |
> | **Multi-Object Editing** | SceneDirector (Ours) | **0.335** | **28.8** |
> | | DriveEditor | 0.376 | 26.9 |
> | | VACE | 0.374 | 27.3 |
>
> SceneDirector achieves the best background preservation across all settings. In the single-object setting, we outperform DriveEditor (+0.9 dB PSNR). This advantage further widens in multi-object editing, as our single-pass approach prevents the accumulated background degradation seen in sequential baselines. We will include these metrics in the revision.
>
> ---
> We sincerely thank you for your supportive evaluation and constructive feedback. We will incorporate all discussed clarifications and additional results to strengthen the final manuscript.

---

> > ### Author Rebuttal · Reviewer_DF23 · 2026-04-01
> >
> > Thanks for the authors' response. As my score is already on the higher side, I will maintain my score.

---

> > > ### Author Response · Authors · 2026-04-03
> > >
> > > Thank you very much for reviewing our rebuttal and for your continued strong support of our work. We appreciate your suggestions, which have helped make our evaluation more comprehensive. We will include the additional metrics and clarifications in the revised manuscript. Thank you again for your encouraging evaluation!

---

### Official Review · Reviewer_J3dM · 2026-03-12

**Soundness:** 3
**Presentation:** 3
**Significance:** 3
**Originality:** 3
**Overall Recommendation:** 4
**Confidence:** 2

**Summary:**

This paper studies autonomous driving video scene editing and aims to support both local object editing and global ego-trajectory editing within a unified framework. Methodologically, the authors use a unified geometric scaffold to provide structural constraints, a texture reference to provide appearance priors, and a gated attention mechanism based on geometric uncertainty to fuse the two, thereby balancing controllability and generation quality. Overall, the problem setting is meaningful, the motivation is clear, and the experimental results are fairly convincing.

**Compliance With Llm Reviewing Policy:**

Affirmed.

**Key Questions For Authors:**

Since the method is built on top of a strong video generation backbone(cosmos-transfer), it is still not entirely clear how much of the improvement comes from the proposed method itself versus the backbone.

**Limitations:**

Yes

**Strengths And Weaknesses:**

## Strengths

1. **The task setting is valuable**
   Handling object editing and trajectory editing simultaneously in a unified framework is practically useful for autonomous driving simulation and data augmentation.

2. **The method is fairly complete**
   From data construction to geometric representation and generative fusion, the overall pipeline is coherent rather than a simple stacking of modules.

3. **The core module shows some novelty**
   Using geometric uncertainty to modulate the strength of texture injection is a reasonable design and fits well with the fact that structural reliability is uneven across driving scenes.

4. **The empirical findings are broadly consistent with the method’s intuition**
   The reported qualitative results and ablation findings generally align with the authors’ motivation, which makes the paper reasonably convincing.

## Weaknesses

1. **A strong backbone may account for a substantial portion of the gains**
   Since the method is built on top of a strong video generation backbone, it is still not entirely clear how much of the improvement comes from the proposed method itself versus the backbone.

2. **The overall system is fairly complex**
   The method relies on multiple components and a relatively heavy geometric/data processing pipeline, which may make reproduction and deployment costly.

---

> ### Author Rebuttal · Authors · 2026-03-31
>
> Thank you sincerely for your positive and encouraging evaluation of our work. We are glad you found our unified task setting *"valuable"* and *"practically useful."* We also appreciate your recognition of our pipeline as *"coherent rather than a simple stacking of modules,"* our uncertainty-guided core module as showing *"some novelty,"* and our empirical findings as *"broadly consistent"* with our intuition.
>
> These observations resonate with the other reviewers. Reviewer SBa5 praised the framework as *"well-motivated and technically robust,"* while Reviewer DF23 highlighted the *"compelling evidence"* of its effectiveness, calling it a *"solid contribution."* Collectively, the reviewers affirm:
> 1.  The practical importance of unifying object and trajectory editing.
> 2.  The principled and coherent design of our multi-component pipeline.
> 3.  The strength and validity of our experimental results.
>
> We are encouraged by this positive consensus and address your specific concerns below.
>
> ---
> ## Q1: A strong backbone may account for a substantial portion of the gains.
>
> This is an important question. We provide multiple lines of evidence that the proposed components—not the backbone alone—are the primary drivers of SceneDirector's performance:
> 1. **Ablation study (Table 4):** Removing our Mask-Gated Reference Attention (w/o Ref. Attn) while **keeping the same backbone** leads to significant degradation: FVD increases by +25.8 (3m Deviation) and +38.4 (2m Offset). Replacing MGRA with standard (ungated) cross-attention still degrades FVD by +12.6 and +19.3, respectively. These results isolate the contribution of our architectural innovations from the backbone.
> 2. **Comparison against a stronger backbone:** VACE-14B uses a substantially **larger 14B-parameter backbone**, yet SceneDirector (built on the smaller Cosmos-Transfer2.5) outperforms it across nearly all metrics. In Multi-Edit: FID 38.29 vs. 49.18, FVD 516.83 vs. 729.42, ATE 0.81m vs. 1.09m (Table 1). This directly demonstrates that our method's gains are not attributable to backbone scale.
> 3. **The backbone cannot perform unified editing alone.** Cosmos-Transfer2.5 is a general-purpose video world model that accepts structural conditions but has no mechanism for: (a) constructing scene-specific geometric scaffolds, (b) integrating editable 3D assets, (c) bridging geometric guidance with texture reference via uncertainty-aware gating, or (d) performing unified object + trajectory editing. These capabilities are entirely enabled by our proposed framework.
> 4. **Enhancement of reconstruction pipelines:** SceneDirector+SG (Tables 2 and 3) outperforms StreetCrafter in Gradual Transition (FID 25.70 vs. 26.36, F1 60.4 vs. 59.5), demonstrating that our generated outputs provide high-quality priors that improve even external reconstruction methods—a benefit attributable to our framework, not the backbone.
>
> We will add a more explicit discussion disentangling the backbone's contributions from those of our method in the revised manuscript.
>
> ---
> ## Q2: System complexity and reproducibility.
>
> We appreciate this practical concern. While our task—unifying multi-view, multi-object editing and trajectory control in a single pass—is inherently complex, we clarify that **the data processing pipeline is an offline process used only for generating training data**. In reality, our scaffold construction is training-free and our inference is highly efficient, making **our method significantly less costly than existing baselines**:
> 1. **Inference efficiency (Appendix F, Table 7):** SceneDirector efficiently edits 4 objects across 5 views simultaneously at $1280 \times 704$ resolution in ~15 minutes. To edit 4 objects in just a single view, baselines require sequential processing: VACE takes ~32 mins ($830 \times 480$) and DriveEditor takes ~20 mins ($1024 \times 576$).
> 2. **Training-free scaffold construction:** Our Unified Geometric Scaffold is training-free. It uses off-the-shelf depth models and standard point cloud operations, making construction drastically faster than reconstruction baselines like StreetGaussian (\~40 mins) and StreetCrafter (\~60 mins), which require costly per-scene optimization.
> 3. **Reproducible methodology:** We guarantee reproducibility by detailing full network/training hyperparameters (App. A), formalized automated data generation pipelines (App. C/D), and explicitly defined evaluation protocols (Sec. 4).
>
> This complexity is inherent to the task scope—simultaneously handling multi-object editing across multiple views with trajectory control. Furthermore, our modular design ensures each component can be substituted or improved independently.
>
> ---
> We hope these responses adequately address your concerns. We believe the evidence presented demonstrates that our proposed components contribute substantially beyond the backbone, and that the system's complexity is well-motivated and manageable. Thank you again for your constructive and supportive review.

---

> > ### Author Rebuttal · Reviewer_J3dM · 2026-04-03
> >
> > Thank you! The review have resolved all my concerns.

---

> > > ### Author Response · Authors · 2026-04-03
> > >
> > > Thank you for taking the time to read our rebuttal and for the positive update. Your practical feedback is highly valuable, and we will integrate the extended discussions from our rebuttal into the final manuscript to further strengthen the paper. Thank you again for your valuable insights and guidance!

---

### Official Review · Reviewer_SBa5 · 2026-03-13

**Soundness:** 2
**Presentation:** 2
**Significance:** 2
**Originality:** 2
**Overall Recommendation:** 4
**Confidence:** 3

**Summary:**

The paper introduces SceneDirector, a novel framework that unifies object editing (insertion, deletion, replacement, repositioning) and ego-trajectory editing (viewpoint control) in driving scene videos within a single inference pass. The framework achieves this by integrating a Unified Geometric Scaffold for structural guidance and a Static Texture Bank for appearance context. These components are harmonized by a Mask-Gated Reference Attention mechanism, which preserves structural integrity while synthesizing fine details in semantically complex regions. Extensive experiments demonstrate the method's superior controllability and high generation quality.

**Compliance With Llm Reviewing Policy:**

Affirmed.

**Final Justification:**

My concerns have been addressed in rebuttal and I will keep my ratings.

**Key Questions For Authors:**

See Weakness

**Limitations:**

yes

**Strengths And Weaknesses:**

Strengths:
1. The proposed framework is both well-motivated and technically robust. It addresses a critical and underexplored challenge in driving scene editing, the reconciliation of geometric precision for trajectory control with generative flexibility for object manipulation, through a thoughtfully designed architecture.
2. The paper is well-written and clearly structured. The motivation, methodology, and experimental setup are presented in a logical and accessible manner, making it easy for readers to grasp both the high-level concepts and the technical details.
3. The paper addresses an important and relevant problem in autonomous driving: the need for realistic and controllable data generation for validating perception systems.

Weakness:
1. While the framework effectively integrates several existing techniques, its core novelty lies more in the thoughtful combination and adaptation of these components rather than in the introduction of entirely new methodologies.
2. The use of synthetically perturbed data for self-supervised training introduces a domain gap between the training distribution and real-world captures, which could potentially compromise the model's robustness in practical applications.
3. In Section 3.1, line 149, the formulation of the Rectified Flow appears to be incorrectly stated. The standard formulation should be $x_t=tx+(1-t)\epsilon$ where $x$ is the data point and $\epsilon$ is the noise. The current equation may contain a typo.
4. Due to the absence of an explicit 3D representation in the generated outputs, the results exhibit certain deficiencies in spatio-temporal consistency.

---

> ### Author Rebuttal · Authors · 2026-03-31
>
> We deeply appreciate your constructive evaluation and recognition that our framework addresses a *"critical and underexplored challenge."* We are thrilled that you found the methodology *"well-motivated and technically robust."*
>
> These positive observations resonate across all reviews, affirming three key strengths:
> 1.  The practical importance of unifying object and trajectory editing.
> 2.  The principled and coherent design of our multi-component pipeline.
> 3.  The strength and validity of our experimental results.
>
> We address your specific questions below:
>
> ---
> ## Q1: Novelty lies in combination rather than entirely new methodologies.
> Thank you for your feedback. Reconciling generative flexibility (for realistic object editing) with physical precision (for trajectory control) remains a critical challenge in driving scene editing. Rather than concatenating tools, our core novelty lies in proposing a new **system-level paradigm** that bridges explicit geometry and generative priors to achieve unified editing:
> 1. **Mask-Gated Reference Attention (MGRA):** Existing methods struggle to balance strict 3D consistency with photorealistic hallucination in occluded regions. We introduce a novel uncertainty-aware fusion that balances physical constraints and hallucination, solving the ghosting artifacts inherent in naive feature fusion.
> 2. **Unified Geometric Scaffold:** Current trajectory editing relies heavily on costly per-scene optimization (e.g., NeRF/3DGS). We shift this paradigm by proposing a training-free, explicit 3D scaffold. This innovation provides a unified spatial anchor for both object and ego-trajectory edits simultaneously, advancing scalability and efficiency.
> 3. **Self-supervised pair construction pipeline:** We solve the paired-data bottleneck by coupling tailored view perturbations (for trajectory shifts) with masked 3D asset injection (for object manipulation).
>
> We view this holistic architecture as a fundamental system-level innovation that unlocks unified, single-pass scene editing. We will emphasize this in the revision.
>
> ---
> ## Q2: Domain gap from synthetically perturbed training data.
>
> We agree that synthetic data may introduce a domain gap. However, capturing paired real-world data for driving scene edits is impossible. Consequently, constructing self-supervised pairs via synthetic perturbation is **a standard paradigm in the driving video editing field**.
>
> While we follow this established practice, we actively minimize the domain gap by introducing physics-grounded improvements. Rather than naive image augmentations, our engineered perturbations utilize kinematic-aware curves and view-dependent parallax transforms to strictly simulate authentic real-world maneuvers, such as realistic accelerations and lane changes. **For a detailed breakdown of these specific perturbation mechanisms and how they compare to prior art, please refer to our response to Reviewer DF23's Q1.**
>
>
> ---
> ## Q3: Rectified Flow formulation may contain a typo.
>
> We appreciate your careful scrutiny. **Both formulations are correct**, differing only in boundary conventions for time $t$:
>
> * **Our formulation (Sec. 3.1):** $x_t=(1-t)x+t\epsilon$. Following our backbone Cosmos-Transfer2.5 (Ali et al., 2025; Eq. 1-3), $t=0$ is clean data $x$ and $t=1$ is noise $\epsilon$. This yields the velocity field $u_t=\epsilon-x$, which aligns with our loss function in Equation (1).
> * **Your formulation:** $x_t=tx+(1-t)\epsilon$, where $t=0$ represents noise and $t=1$ represents data, reflecting the original Rectified Flow paper (Liu et al., 2023).
>
> We apologize for the confusion and will **explicitly state our time convention** in the revised Section 3.1.
>
> ---
> ## Q4: Spatio-temporal consistency without explicit 3D representation.
>
> Thank you for this observation. While our final output is 2D, we explicitly enforce strict spatio-temporal coherence by these **architectural guarantees**:
> 1. **Unified Geometric Scaffold** anchors a globally consistent 3D point cloud, rendering it across all target views via a pinhole camera model.
> 2. **Static Texture Bank** acts as a frozen, shared memory to guarantee globally consistent appearance cues.
> 3. **Temporal Backbone** processes 3D latents with causal attention to naturally enforce temporal coherence.
>
> **Empirical evidence confirms strong real-world robustness:**
> 1. **Cross-View Stability:** Under large viewpoint shifts, our DINOv2 structural consistency (0.789) nearly matches Real Data (0.798), eliminating geometric drift in baselines like FreeVS (0.683).
> 2. **3DGS Convergence:** Training StreetGaussian on our multi-view outputs achieves SOTA reconstruction (FID 25.70). Because 3DGS only converges under strict multi-view consensus, this proves our strong spatio-temporal coherence.
>
> ---
> We truly value the rigorous scrutiny you applied to our work. We hope these responses address your concerns. We are committed to adding the suggested revisions and believe they will further strengthen the paper.

---

> > ### Author Rebuttal · Reviewer_SBa5 · 2026-04-01
> >
> > My concerns have been addressed and I will keep my ratings.

---

> > > ### Author Response · Authors · 2026-04-03
> > >
> > > Thank you for your time. We appreciate your rigorous review and constructive feedback. As promised, we will ensure that all the clarifications and details provided in our rebuttal are incorporated into the final version of the manuscript. Thank you again for your support of our work!

---

### Decision · Program_Chairs · 2026-04-30

**Decision:**

Accept (regular)

**Comment:**

This paper received all positive reviews. Reviewers agree that the work is valuable for driving scene video editing which also makes contribution to video world model research. All reviewers stated that author's rebuttal fully resolved their concerns.
The spatio-temporal consistency without an explicit 3D representation is worth discussing.